# Privacy Risks and Preservation Methods in Explainable Artificial Intelligence: A Scoping Review

**Sonal Allana**                                           *sallana@uoguelph.ca*
*School of Computer Science*
*University of Guelph*

**Mohan Kankanhalli**                                      *mohan@comp.nus.edu.sg*
*School of Computing*
*National University of Singapore*

**Rozita Dara**                                            *drozita@uoguelph.ca*
*School of Computer Science*
*University of Guelph*

**Reviewed on OpenReview:** *https://openreview.net/forum?id=q9nykJfzku*

## Abstract

Explainable Artificial Intelligence (XAI) has emerged as a pillar of Trustworthy AI and aims to bring transparency in complex models that are opaque by nature. Despite the benefits of incorporating explanations in models, an urgent need is found in addressing the privacy concerns of providing this additional information to end users. In this article, we conduct a scoping review of existing literature to elicit details on the conflict between privacy and explainability. Using the standard methodology for scoping review, we extracted 57 articles from 1,943 studies published from January 2019 to December 2024. The review addresses 3 research questions to present readers with more understanding of the topic: (1) what are the privacy risks of releasing explanations in AI systems? (2) what current methods have researchers employed to achieve privacy preservation in XAI systems? (3) what constitutes a privacy preserving explanation? Based on the knowledge synthesized from the selected studies, we categorize the privacy risks and preservation methods in XAI and propose the characteristics of privacy preserving explanations to aid researchers and practitioners in understanding the requirements of XAI that is privacy compliant. Lastly, we identify the challenges in balancing privacy with other system desiderata and provide recommendations for achieving privacy preserving XAI. We expect that this review will shed light on the complex relationship of privacy and explainability, both being the fundamental principles of Trustworthy AI.

## 1 Introduction

### 1.1 Paradigm shift in technology and the need for explanations

Traditional software development processes have metamorphosed into stable and reliable frameworks through decades of fine tuning by software experts. These software systems are built on human designed algorithms and produce a trace of the logic used to generate an output. Even in complex systems, it is possible for software experts to analyze the logic and generate an explanation for a specific result. During the software development lifecycle, engineers focus on creating the algorithm and validating using well designed test cases that closely replicate real world scenarios. In contrast, modern AI systems do not have an underlying human-written algorithm and learn from data fed to them. This data-driven nature creates dependence of the system on data quality (Merhi, 2022) and introduces problems such as lack of fairness when data

is biased, or irrelevant results when data is incomplete or outdated (Trocin et al., 2021). During the AI development phase, engineers access training datasets which may contain personally identifiable or sensitive information about individuals. For neural network systems, the development process often involves a trial-and-error approach, where high accuracy is targeted by tweaking the hyperparameters such as the learning rate, epochs, number of layers or activation functions. The lack of an algorithm prevents engineers from tracing through the AI system and interpreting the results. Thus, the basic ability to be explainable and understand input-output behaviors, which is critical to all computer systems (Sundararajan et al., 2017), is often out of reach of AI systems. Explanations for outcomes of AI are crucial in high-risk applications (Mochaourab et al., 2023) in domains such as healthcare (Dhar et al., 2023; Dwivedi et al., 2023; Yang et al., 2022), finance (Dwivedi et al., 2023; Zhang et al., 2022), defense (Dwivedi et al., 2023), justice (Deeks, 2019), energy and power (Machlev et al., 2022) where the impact on human life and well-being is significant (Karimi et al., 2023; McDermid et al., 2021; Nassar et al., 2020) and the inability to do so deters their successful implementation (Nassar et al., 2020; Shrikumar et al., 2017a; Yang et al., 2022).

Trustworthy AI strives to mitigate risks due to possible harms from the data-driven nature of AI systems. Trustworthiness is based on foundation principles of reliability, validity, robustness, privacy, explainability and fairness (Alzubaidi et al., 2023; Tabassi, 2023) to boost user confidence in the system outputs. Among these principles, explainability aims to bring the much-needed transparency in opaque models and can be considered as a non-functional requirement of a software system to mitigate opacity (Chazette et al., 2021). There are numerous benefits of including explanations in AI models. Besides aiding data scientists in getting a better understanding of the data (Hohman et al., 2019) and performing required data cleansing (Chen et al., 2022b), explanations can help developers in detecting errors in input and determining features that can be modified to change the outcome (Datta et al., 2016). When multiple models are available with similar accuracy, an explanation method can help to choose between models (Dhurandhar et al., 2018). Interpretable models can enable knowledge discovery by detecting knowledge or patterns that were missed by uninterpretable ones (Kim et al., 2016). Since humans remain an important component in the decision-making process as end-users and consumers of automated decisions (Terziyan & Vitko, 2022), explanations can give them an understanding of the model outcome, especially when they are adversely affected by the decisions (Ali et al., 2023). Explainability can also facilitate privacy awareness in end-users (Brunotte et al., 2021), enabling them to make right choices for their personal data and aid regulators and compliance officers to understand the compliance of models (McDermid et al., 2021) with applicable regulations. With generative AI (Gen-AI) and large language models (LLMs) entering mainstream, explanations constitute an important design principle (Weisz et al., 2023) in enabling a better mental model for users (Sun et al., 2022) and in communicating its capabilities and limitations to them (Weisz et al., 2023). It can also support users in effective prompt engineering to determine the words that impact the output of a model (Mishra et al., 2023) and in verification of generated content to mitigate the problem of hallucinations (Schneider, 2024b).

## 1.2 Challenges for privacy in explainability

In many high risk application domains of AI, training models on sensitive personal information is inevitable for usefulness of these systems (Veugen et al., 2022). For instance, a lung cancer detection model necessitates training on chest X-ray images, which constitutes personal information of patients. Similarly, a loan evaluation model of a bank, requires access to the financial profiles of customers, which is also personal information of individuals. Usage of personal data impacts the privacy of individuals when they are subject to intentional or unintentional identification and exposure through these systems. Some models are found to memorize data contained in the input (Song et al., 2017) which can be exploited by adversaries for extraction of personal information. Gen-AI models create new content from large multi-modal datasets (Sun et al., 2022) which could potentially contain sensitive personal information (Meskó & Topol, 2023). Due to such privacy risks involved, when personal data is used in training, testing, or inferencing of AI models, they become subject to data regulation and privacy acts (ICO, 2020).

Explainability is a foundational principle of Trustworthy AI, however, recent research has determined that introducing explanations in AI systems is found to conflict with the privacy requirements of the system. Explanation interfaces are found to give adversaries an additional attack surface (Duddu & Boutet, 2022; Liu et al., 2024) to mine the information contained in the model. Privacy attacks can target explanations

to retrieve information about membership in the training set (Liu et al., 2024; Naretto et al., 2022; Shokri et al., 2021), build surrogates of the underlying model (Aïvodji et al., 2020; Wang et al., 2022; Yan et al., 2023b), infer sensitive attributes of individuals (Duddu & Boutet, 2022; Luo et al., 2022) and reconstruct the training set (Shokri et al., 2021). This leakage is demonstrated across different types of XAI methods including those that are currently used in commercial production systems. In addition to privacy attacks, the content of explanations may also inadvertently expose information that is proprietary (Milli et al., 2019) and hence valuable and confidential to organizations (Winikoff & Sardelic, 2021) or sensitive to individuals, thus causing breach of data and privacy regulations. Hence researchers have highlighted the urgent need of mitigating privacy leakage through explanation interfaces (Luo et al., 2022; Patel et al., 2022; Yan et al., 2023b). Due to these concerns of the privacy vulnerabilities of explanations, necessary privacy preservation measures are required in XAI systems (Aïvodji et al., 2020; Shokri et al., 2021; Zhao et al., 2021).

### 1.3 Main contributions

Previous research has identified that the privacy issues in explainability are insufficiently studied (Liu et al., 2024; Luo et al., 2022; Naretto et al., 2022) despite its criticality in achieving safety in AI transparency. To the best of our knowledge, there is currently no work that provides an in-depth understanding of the conflict between privacy and explainability in AI. Hence, we focus this article on these two fundamental desiderata of Trustworthy AI and explore the landscape of privacy risks and preservation methods proposed in literature in the context of XAI. The key questions that we have designed to define the scope of this article are:

RQ1: What are the privacy risks of releasing explanations in AI systems?

RQ2: What current methods have researchers employed to achieve privacy preservation in XAI systems?

RQ3: What constitutes a privacy preserving explanation?

We conducted a scoping review guided by RQ1 and RQ2. Based on the knowledge gathered from the extracted studies, we propose characteristics of privacy preserving XAI and outline them with the help of practical use cases to answer RQ3. Our main contributions in this article are as follows:

- *Categorization of reported privacy risks in XAI:* We review the conflict between privacy and explainability in current literature and categorize the risks.

- *Identification of applicable privacy preservation methods in XAI:* We determine the privacy preservation methods that are applicable to XAI and report the progress achieved by researchers in integrating them in XAI systems.

- *Privacy preserving XAI characteristics:* We propose the desirable characteristics of privacy preserving XAI to provide researchers and practitioners the guidelines for achieving the trade-off between privacy, utility and explainability.

The rest of this article is organized as follows. Section 2 presents a brief background on XAI including its definition, evolution, categorization of explanation approaches and related reviews. In Section 3, we present the details of the scoping review methodology for extracting studies relevant to our research questions. Sections 4 and 5 synthesize the results from the scoping review. In Section 4, we consolidate both intentional and unintentional privacy risks of explanations to answer RQ1. In Section 5, we elaborate the use of privacy preserving methods on explanations and the existing works that utilize them in response to RQ2. Section 6 proposes the characteristics of privacy preserving XAI and answers RQ3. We conclude the article by discussing the results, and highlight the open issues, challenges, and recommendations for future work in Section 7 and conclusions in Section 8.

## 2 Background

### 2.1 Definition of XAI

In 2017, DARPA kickstarted its 4-year XAI program to accelerate research in the development of explanation methods and interfaces to enhance understanding and trust of end-users (Gunning & Aha, 2019). The program defined XAI as "AI systems that can explain their rationale to a human user, characterize their strengths and weaknesses, and convey an understanding of how they will behave in the future" (Gunning & Aha, 2019). The study established users' preference for systems with explanations over systems that provided only decisions. Ribeiro et al. (2016) refer to explanations of predictions as qualitative artifacts that provide the relationship between an input instance and the output prediction.

### 2.2 Evolution of XAI and emergence of privacy concerns

The field of explainability can be traced to the early 1990s, driven by the lack of transparency in black-box models. Early contributions (Benitez et al., 1997; Craven & Shavlik, 1995; LiMin Fu, 1994; Milaré et al., 2002; Torres & Rocco, 2005) proposed different techniques for extracting interpretable representations from these systems. The rise of deep learning and the improvement in the predictive performance of black-box systems, propelled complex uninterpretable systems into mainstream usage. However, their use in critical domains remains problematic due to their lack of transparency. Regulatory frameworks such as the General Data Protection Regulation (GDPR, 2016), specifically the provisions on individuals' rights related to automated decision-making including profiling, intensified the demand for transparent, explainable models thus resulting in a rapid growth in the field of explainability.

However, the introduction of transparency through XAI methods has also exposed new vectors for privacy leakage through explanation interfaces. Early studies (Milli et al., 2019; Shokri et al., 2020; 2021) described privacy attacks on the training data and the underlying model. In response, researchers have begun to explore defense mechanisms and pioneering works in this field (Harder et al., 2020; Patel et al., 2022) have proposed various strategies for generating privacy preserved explanations. Despite these efforts, privacy risks in XAI remain an open research problem, with novel attacks being identified and defense strategies being actively investigated. Figure 1 outlines the key milestones in the evolution of XAI and highlights the emergence of privacy issues and proposed defenses.

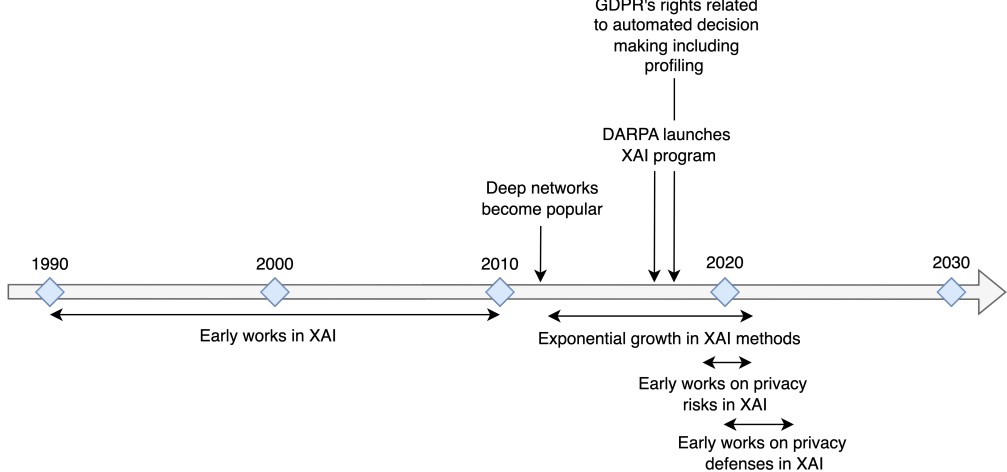

Figure 1: Key milestones and emergence of privacy attacks/defenses in XAI.

### 2.3 Categorization of XAI

In recent years, several different XAI methods have been proposed. Broadly, explainability can be achieved using inherently interpretable models or applying post-hoc methods on trained models (Harder et al., 2020).

Methods specific to certain model types and capabilities, are referred to as model-specific while those independent of the model are referred to as model-agnostic (Dwivedi et al., 2023). In this subsection, we discuss the main categories into which XAI methods are grouped in existing literature (Table 1), based on the underlying mechanism used to derive explanations. We also include categories suitable for Gen-AI (Sections 2.3.5 to 2.3.8) that explore new explanation paradigms for large models (Sarkar et al., 2024). Since there is a broad spectrum of available explainability methods, we limit ourselves to a selection of methods to give readers sufficient understanding of the terminologies used in subsequent sections. For a comprehensive review of XAI categories and methods, we refer the reader to other related reviews listed in Section 2.4.

### 2.3.1 Interpretable methods

These AI models are understandable by design (Arrieta et al., 2020). They have embedded rules or transparent architecture that facilitates the understanding of the input-output logic of the system. They are also referred to as white-box or transparent models. Decision trees, Bayesian networks, linear/logistic regression, k-nearest neighbours, rule based systems and general additive models (Arrieta et al., 2020; Molnar, 2023; Rawal et al., 2022) are some examples of interpretable models. Though these models are promising in aiding the understandability of a system, they have limitations. The primary deterrent to their successful adoption as explainable-by-design methods, is their lower accuracy (Blanco-Justicia et al., 2020; El Zein et al., 2024; Gunning & Aha, 2019) compared to better performing black-box models such as deep learning systems. They also lack natural language explanations, making them unsuitable for use by non-technical users (Biran & McKeown, 2017). Nonetheless, due to their intrinsically transparent architecture, interpretable models are often used as surrogates for black-box models (McDermid et al., 2021). The use of multiple surrogate models is found to facilitate the availability of different types of explanations (Dwivedi et al., 2023) improving the overall interpretability of the system.

### 2.3.2 Example-based methods

These methods use examples, i.e., data instances as samples to explain the model (McDermid et al., 2021). The instances may be from the training set or generated by the method (Jiménez-Luna et al., 2020; Li et al., 2020b). These methods are also referred to as record-based (Shokri et al., 2020), instance-based (Jiménez-Luna et al., 2020) or case-based (Montenegro et al., 2021) methods in literature. They can complement feature-based methods to aid understandability of the end users (Jia et al., 2021) and also improve the interpretability of complex distributions (Kim et al., 2016). They are intuitive and natural in their ability to provide explanations to humans (Jiménez-Luna et al., 2020). Some methods in this category are anchors (Ribeiro et al., 2018), contrastive explanations (Dhurandhar et al., 2018), counterfactuals (Wachter et al., 2017), influence functions (Koh & Liang, 2017) and prototypes and criticisms (Kim et al., 2016).

### 2.3.3 Knowledge-based methods

These methods utilize knowledge representation techniques in machine learning (ML) models to enhance interpretability (Tiddi & Schlobach, 2022). The integration of background knowledge (Hitzler et al., 2022) facilitates the incorporation of contextual information (Lecue, 2020; Páez, 2019), thus increasing the trustworthiness of explanations. The emerging field of neuro-symbolic (Hitzler et al., 2022) or in-between methods (Ilkou & Koutraki, 2020) explores the integration of symbolic AI approaches rooted in knowledge representation and reasoning with subsymbolic or connectionist based approaches (Hitzler et al., 2020).
Another knowledge-based approach is the use of semantic web technologies for semantic interpretation and automated reasoning from structured knowledge bases (Seeliger et al., 2019). Knowledge graphs and ontologies are the common tools that can be deployed to support explainability. Knowledge graphs have applicability in pre-model and post-model explainabilty contexts for feature extraction, relation identification, inferencing and reasoning (Rajabi & Etminani, 2022). The field of semantic web technologies in explainability is attractive because of its potential in creating knowledge-rich explanations without compromising the model performance (Seeliger et al., 2019).

### 2.3.4 Feature-based methods

These explanation methods score or measure the effect of individual input features on the output of the model (Arrieta et al., 2020; Bhatt et al., 2020; Dwivedi et al., 2023; Strobel & Shokri, 2022). They are also referred to as feature importance (McDermid et al., 2021), feature relevance (Arrieta et al., 2020) or attribution-based (Liu et al., 2024) methods. They are based on the attribution problem which is the distribution of the output of a model for a specific input to its base features (Sundararajan & Najmi, 2020). Two important categories of feature-based methods identified in literature are perturbation and backpropagation-based methods (Ancona et al., 2018; McDermid et al., 2021).

- *Perturbation-based methods* remove, alter, or mask an input feature or set of features and observe the difference with the original output (McDermid et al., 2021). Some perturbation-based methods are LIME (Ribeiro et al., 2016), permutation feature importance (Breiman, 2001), SHAP (Lundberg & Lee, 2017) and MASK (Fong & Vedaldi, 2017).

- *Backpropagation-based methods* compute input attributions in forward and backward passes of the network (Ancona et al., 2018). The use of the gradient of the output with the respective input features (McDermid et al., 2021; Strobel & Shokri, 2022) is a common approach in these methods and is referred to as gradient-based approach. Methods used on images that determine the global importance of pixels, generate saliency maps, and are referred to as pixel-level attribution methods (Kapishnikov et al., 2019; Molnar, 2023). Some examples of backpropagation-based methods are gradient (Simonyan et al., 2014), gradient x input (Shrikumar et al., 2017b), guided backpropagation (Springenberg et al., 2015) and integrated gradients (Sundararajan et al., 2017).

### 2.3.5 Concept-based methods

Concept-based methods aim to uncover high-level concepts above features, pixels and characters (Ghorbani et al., 2019), that influence the behaviour of a model (Bereska & Gavves, 2024). The use of human understandable concepts result in naturally interpretable explanations. Desiderata such as meaningfulness, coherence and importance are proposed for this category of explanations (Ghorbani et al., 2019). Concept activation vectors (CAVs) (Kim et al., 2018) is a concept-based method that leverages linear classifiers to determine the presence or absence of concepts corresponding to a set of input examples. The use of concept activation regions (Crabbé & van der Schaar, 2022) enhances CAVs and allows the underlying examples to be distributed across the latent space of the model, resulting in the discovery of global explanations. In another approach, concept bottleneck models (Koh et al., 2020) predict intermediate human-specified concepts in the model and utilise these for making predictions. The models also support human interventions on these concepts and facilitate the generation of counterfactuals, thus enabling human-model collaboration (Koh et al., 2020).

### 2.3.6 Probing-based methods

Probing-based methods use classifiers or probes to determine the knowledge captured by a model (Sajjad et al., 2022; Schneider, 2024a). The technique involves using the internal activations of networks to train probing classifiers for prediction of embedded properties (Alain & Bengio, 2017; Belinkov, 2022). However, such probing experiments require the properties to be known apriori along with clear specifications of the original/probing tasks, the models and datasets used. The outcomes of these experiments can be applied for tuning the model on certain tasks or determining the encoded information for downstream tasks.

### 2.3.7 Neuron activation methods

These methods focus on the behaviour of neurons that are responsible for specific outcomes or represent learned linguistic properties (Zhao et al., 2024). Proposed methods in this category, such as linguistic correlation analysis and cross-model correlation analysis, identify and analyse such neurons for post-hoc explainability (Dalvi et al., 2019). Individual or groups of neurons may be discovered as salient for a property and ranked based their importance to the model's task (Dalvi et al., 2019). These methods may also experience trade-offs between accuracy and selectivity (Antverg & Belinkov, 2022).

### 2.3.8 Mechanistic interpretability

Mechanistic interpretability (Bereska & Gavves, 2024) seeks to understand the model outputs by reverse engineering (Olah, 2022) the system. Analogous to understanding complex computer programs, this approach aims at dissecting network activations into independently understandable units. It considers the analysis of individual components of the system, such as features, connections and neurons, to determine the causes of outputs. This approach to interpretability includes techniques that could be applied before, during or after the training process. Techniques, such as feature visualisation (Zimmermann et al., 2021) and sparse autoencoders (Cunningham et al., 2024), are some of the methods from this category.

**Table 1** Broad XAI categories and a selection of early works.

| XAI Category | XAI Method | | Model-specific/agnostic | Study |
|---|---|---|---|---|
| Interpretable | Decision trees, Bayesian networks, linear/logistic regression, k-nearest neighbours, rule-based systems, general additive models | | Model-specific | - |
| Example-based | Anchors | | Model-agnostic | Ribeiro et al. (2018) |
| | Contrastive explanations | | Model-agnostic | Dhurandhar et al. (2018) |
| | Counterfactuals | | Model-agnostic | Wachter et al. (2017) |
| | Influence functions | | Model-agnostic | Koh & Liang (2017) |
| | Prototypes and criticisms | | Model-agnostic | Kim et al. (2016) |
| Knowledge-based | Semantic web technologies | | Model-agnostic | Seeliger et al. (2019) |
| | Neuro-symbolic approaches | | Model-specific | Hitzler et al. (2022) |
| Feature-based | Perturbation-based | LIME | Model-agnostic | Ribeiro et al. (2016) |
| | | Permutation Feature Importance | Model-agnostic | Breiman (2001) |
| | | SHAP | Model-agnostic | Lundberg & Lee (2017) |
| | | MASK | Model-agnostic | Fong & Vedaldi (2017) |
| | Backpropagation-based | Gradient | Model-specific | Simonyan et al. (2014) |
| | | Gradient x Input | Model-specific | Shrikumar et al. (2017b) |
| | | Guided Backpropagation | Model-specific | Springenberg et al. (2015) |
| | | Integrated Gradients | Model-specific | Sundararajan et al. (2017) |
| Concept-based | Concept activation vectors | | Model-specific | Kim et al. (2018) |
| | Concept bottleneck models | | Model-specific | Koh et al. (2020) |
| | Concept activation regions | | Model-specific | Crabbé & van der Schaar (2022) |
| Probing-based | Probing experiments | | Model-specific | Alain & Bengio (2017) |
| Neuron activation | Linguistic correlation analysis | | Model-specific | Dalvi et al. (2019) |
| | Cross-model correlation analysis | | Model-specific | Dalvi et al. (2019) |

| XAI Category | XAI Method | Model-specific/agnostic | Study |
|---|---|---|---|
| Mechanistic interpretability | Feature visualisation | Model-specific | Zimmermann et al. (2021) |
| | Sparse autoencoders | Model-specific | Cunningham et al. (2024) |

## 2.4 Related reviews

XAI is currently an active research area and detailed reviews have captured the state of the art in the field. Though current literature has reviews covering different aspects of XAI, to the best of our knowledge there is a lack of comprehensive review that considers the tension of privacy with explainability. Our work addresses this gap and offers a unique contribution compared to other existing reviews. In this subsection, we identify related reviews on XAI and summarize their focus areas.

An in-depth overview of the core concepts and taxonomies in XAI was provided by Arrieta et al. (2020). Mohseni et al. (2021) conducted an interdisciplinary survey and proposed a comprehensive framework for design and evaluation of XAI methods. Dwivedi et al. (2023) covered a wide breadth of explanation algorithms, programming frameworks and software toolkits for XAI development. Ali et al. (2023) examined explainability through the lens of trustworthiness detailing evaluation metrics, available software packages and XAI datasets. Bodria et al. (2023) systematically categorized explanation methods and benchmarked prominent methods using quantitative metrics. Muralidhar et al. (2023) reviewed transparency elements from human computer interaction (HCI) in the context of explanations while Cambria et al. (2023) investigated presentation methods and usage of natural language with XAI.

Beyond these broad surveys, domain specific reviews have also emerged. For example, XAI in healthcare has been surveyed by Payrovnaziri et al. (2020) and Yang et al. (2022); in cybersecurity by Capuano et al. (2022) and in energy and power systems by Machlev et al. (2022). Methodology focussed reviews also exist, covering counterfactuals (Guidotti, 2022), data-driven knowledge-aware XAI systems (Li et al., 2020b), knowledge-graph based XAI (Rajabi & Etminani, 2022; Tiddi & Schlobach, 2022) and semantic web technologies for explanations (Seeliger et al., 2019). Recent advances include the intersection of explainability and federated learning (FL), termed as Federated XAI (Fed-XAI), reviewed by López-Blanco et al. (2023) and categorisation of explanation techniques for transformer-based language models based on training paradigms as surveyed by Zhao et al. (2024). A comprehensive review of mechanistic interpretabililty in LLMs for achieving AI safety was conducted by Bereska & Gavves (2024).

Recent literature has increasingly highlighted the potential of malicious exploitation of XAI interfaces. Baniecki & Biecek (2024) present a survey of adversarial attacks in XAI and discuss corresponding defenses. While their study addresses a specific type of XAI privacy attack, its primary emphasis is on non-privacy attacks including data poisoning, backdoor attacks, model manipulation and attacks on fairness metrics through explanations. A categorisation of XAI-aware attacks was proposed in another recent work (Noppel & Wressnegger, 2024b;a) based on the outputs that are preserved in the attack. The work also categorised attacks based on the scope of explanation alteration and adversary capabilities. However, this work targets the robustness of post-hoc XAI and though the proposed categorisation is relevant to privacy attacks, it does not directly focus on the privacy aspect of XAI safety.

The focus of this review diverges from prior reviews by specifically examining the privacy risks that target the identification and exposure of personal or sensitive information through the misuse of XAI interfaces in AI systems. Further, we review strategies used by researchers in mitigating the privacy leakage in XAI. An early work by Spartalis et al. (2023), provides a short review on XAI privacy risks and applicable defense mechanisms. The review also identifies security risks such as evasion and poisoning through XAI interfaces. Though this work raises awareness for further research in this area, it mainly focuses on early

privacy attacks and discusses limited defense mechanisms. A recent review (Nguyen et al., 2025) explores the intersection of privacy and explainability through attacks and countermeasures, but differs from our review in its methodology and lacks a detailed examination of the core requirements underpinning the design of privacy preserving XAI. Our review employs an established scoping review methodology guided by clearly defined research questions. The resulting taxonomy of XAI privacy risks and corresponding mitigation methods are distilled from the understanding of existing literature across the privacy and XAI communities. This methodology enables a structured and rigorous approach to addressing the research questions through the analysis of the selected studies.

## 3 Method

We conducted a scoping review based on the Preferred Reporting Items for Systematic reviews and Meta-Analyses extension for Scoping Reviews (PRISMA-ScR) (Tricco et al., 2018). This section elaborates the process followed, the proposed taxonomy of XAI privacy risks and the identified research trends.

### 3.1 Literature selection and extraction

A 4-step process was employed comprising of identification, screening, eligibility, and extraction, as illustrated in Figure 2. In the initial step of identification, Elsevier Engineering Village (Engineering Village) search platform was used and the search was conducted on Compendex and Inspec databases. These databases index publications from leading computer science publishers, including IEEE, ACM, Springer and Elsevier. A researcher formulated the search string, using the two main concepts of privacy and explainability, with the help of a librarian. This was applied on the title, subject, and abstract fields. The reseacher worked with the librarian to set the search, inclusion and exclusion criteria as detailed below for reproducibility:

- *Search string:* (privacy OR confidential* OR "membership inference" OR "model inversion" OR "model extract*" OR "model reconstruct*" OR "property inference") AND (explainab* OR explanat* OR interpretab* OR XAI OR recourse OR "transparency report").

- *Period of publication:* January 1, 2019, to December 31, 2024. The start year was chosen based on the seminal works (Milli et al., 2019; Shokri et al., 2020; 2021) published on this topic.

- *Date of most recent search:* Jan 6, 2025

- *Type of publications included:* journal articles, conference articles, book chapters, articles in press.

- *Type of publications excluded:* preprints, unpublished papers, dissertations, books, standards, report chapters, notes, report reviews, editorials, erratum, retracted documents. This criteria was required to ensure the extraction of high quality and rigorously peer reviewed articles.

- *Language:* English

- *Inclusion criteria:* Study should describe at least one privacy risk or privacy preservation method in XAI.

The search results comprising of 3,766 studies were exported from Engineering Village and imported into Covidence (Covidence) review management software by one researcher. During the import process, the software merged duplicate studies from the databases, retaining only unique records. After deduplication, 1,943 studies were forwarded for screening wherein the title and abstract were examined by the researcher to determine relevance to RQ1 or RQ2 while considering the inclusion criteria. Out of 1,943 studies, 69 studies were moved to the next step to determine eligibility wherein the full text of the identified articles were examined with respect to RQ1 and RQ2. This step was conducted by one researcher and independently verified by a second researcher. Since the focus of this article is privacy leakage through XAI interfaces, studies that addressed adversarial XAI (such as backdoor, poisoning, etc.) without overlap on privacy were eliminated. Studies that discussed general privacy issues in ML without XAI context were also eliminated.

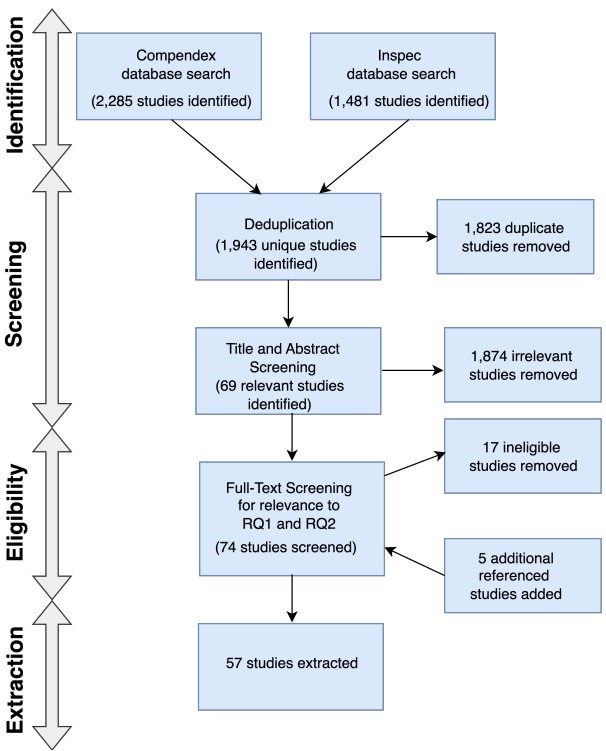

Figure 2: Scoping review process as per PRISMA-ScR.

Related survey papers that appeared in the search results were not extracted but used to determine any additional studies relevant to the research questions. These steps resulted in removal of 17 studies and addition of 5 studies identified through forward and backward searches. The removal and addition of studies were discussed and verified by two researchers. Thus, overall the 4-step scoping review process resulted in extraction of 57 studies.

## 3.2  Proposed taxonomy of privacy risks in XAI

Each extracted study was categorized under the appropriate research question. The categorisation of studies under RQ1, led to the identification of 3 main types of privacy attacks reported in XAI. These attacks are due to the malicious intent of adversaries and we propose to term this leakage as intentional. This leakage can target data (training or query) or models (Figure 3). A fourth type of privacy attack in literature was added to the taxonomy due to its applicability to intentional leakage through explanations, even though it is not currently reported in XAI.

Though all studies extracted under RQ1 belonged to intentional leakage, we note that not all privacy leakage in XAI is due to malicious intent. Hence we propose another category of leakage termed as unintentional. This type of inadvertent leakage can be caused due to training problems, such as overfitting, or design flaws, such as lack of appropriate role-based release of explanations. These problems can be unintentionally introduced in XAI systems and further exploited for intentional leakage by adversaries. In Sections 4.2 and 4.3, we elaborate further on intentional and unintentional leakages respectively.

## 3.3  Research trends

The distribution of the extracted studies for RQ1 (i.e., XAI privacy risks) and RQ2 (i.e., XAI privacy preservation) by year, can be seen in Figure 4(a). An upward trend in the reported privacy risks associated with XAI methods is evident over the period under review. Correspondingly, there has been a noticeable

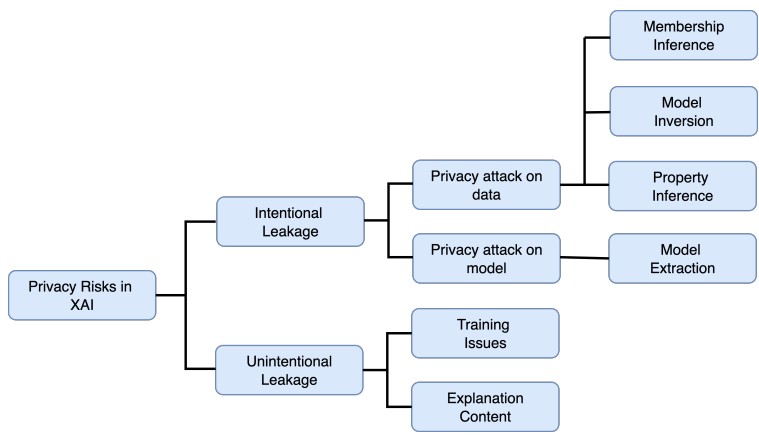

Figure 3: Proposed taxonomy of privacy risks in XAI.

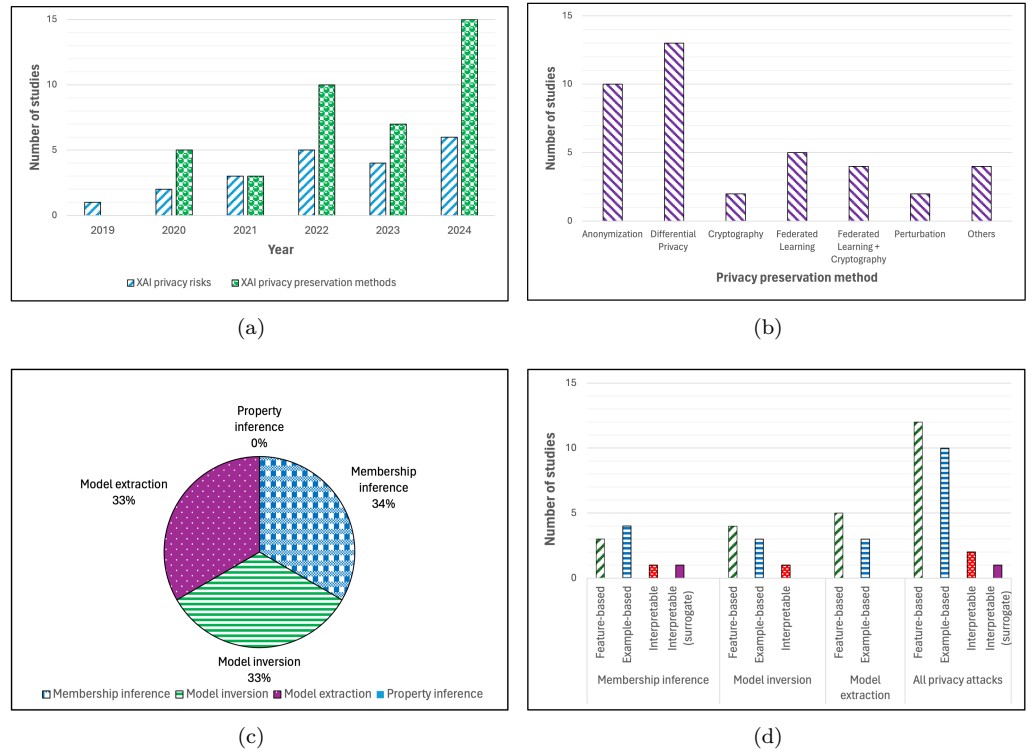

Figure 4: Research trends identified from extracted studies (a) Studies on XAI privacy risks and preservation methods (b) Privacy preservation methods in XAI (c) Privacy attacks in XAI (d) Privacy attacks by XAI categories.

increase in the number of studies exploring the use of various privacy preservation methods in XAI as observed from Figure 4(b). Among these techniques, differential privacy and anonymization emerge as the most commonly employed approaches. With respect to the identified privacy risks, three types of attacks, namely, membership inference, model inversion and model extraction, appear with comparable frequency across literature (Figure 4(c)). Notably, property inference attacks have not been examined in the context of XAI systems. Figure 4(d) presents the categories of XAI targeted by different privacy attacks. Feature-based and example-based XAI are more frequently targeted to such attacks in comparison to interpretable methods.

# 4 Privacy Risks in XAI

Traditionally privacy is referred to as the "right to be left alone" (Warren & Brandeis, 1890) and the "claim of individuals, groups, or institutions to determine for themselves when, how, and to what extent information about them is communicated to others" (Westin, 1967). In the modern context, with availability, collection, and collation of copious information about individuals through online and offline sources, the concept of information privacy is more applicable and refers to the ability of individuals to exert control on their own data (Curzon et al., 2021). Clarke (1999) has defined information privacy as the "claims of individuals that data about themselves should generally not be available to other individuals and organizations, and that, where data is possessed by another party, the individual must be able to exercise a substantial degree of control over that data and its use". In this article, we refer to this latter definition of privacy.

Trustworthy AI is built on the foundational principle of explainability, which supports the gaining of insights into the decision making processes of black-box AI systems (Tabassi, 2023). However, the relationship between privacy and explainability has contrasting aspects. On the one hand, explainability aids privacy in several ways such as in creating privacy awareness in users (Brunotte et al., 2021), in ascertaining that privacy of a system is achieved (Doshi-Velez & Kim, 2017; Müftüoğlu et al., 2022), and in determining correlations with identifiable data for removal (Hohman et al., 2019). On the other hand, explanations can reveal sensitive information contained in models and training data (Harder et al., 2020; Rawal et al., 2022; Zhao et al., 2021) thus leading to privacy risks (Kuppa & Le-Khac, 2021). Thus, there are conflicting outcomes (Guerra-Manzanares et al., 2023; Nguyen et al., 2023; Sanderson et al., 2023; Spartalis et al., 2023) of including explainability as a non-functional requirement in AI systems.

## 4.1 Threat model

To discuss the threat model of XAI, we consider a target AI model, $f$, with a corresponding explanation function, $\phi$. For an input query, $x$, the system generates an output, $f(x)$, such as prediction, classification or an artifact of a Gen-AI system, and a corresponding explanation, $\phi(x)$ (Figure 5). Though the explanation interface is made available for the end-users of the system, adversaries can also secure different levels of access to the XAI system during the stages of the AI lifecycle (Shahriar et al., 2023). In white-box access, adversaries possess information about the model internals such as architecture and hyperparameters (Liu et al., 2022a; Zhang et al., 2023). In black-box access, adversaries access the input/output of the system (Liu et al., 2022a) without any knowledge of the training process or model internals (Rigaki & Garcia, 2023). Any intermediate access between white-box and black-box is referred to as gray-box (Jegorova et al., 2022). The adversaries may act as passive observers (Jegorova et al., 2022) and use the model outputs for launching privacy attacks, or they may actively interfere in the training process of the model (Nasr et al., 2019).

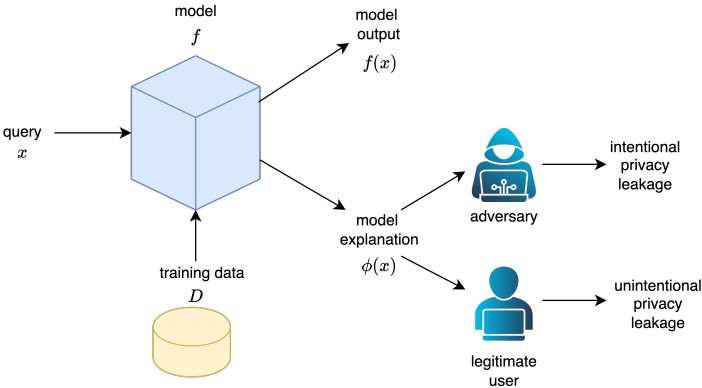

Figure 5: Threat model of XAI.

## 4.2 Intentional privacy leakage

This subsection reviews intentional risks in the form of privacy attacks launched by adversaries. As XAI systems are fundamentally AI models augmented with explainability features, they remain susceptible to malicious threats that affect conventional AI models. Prior research has identified security attacks, such as evasion and poisoning (Pitropakis et al., 2019), that compromise the integrity of AI. Adversarial attacks on XAI such as input manipulations (Dombrowski et al., 2019; Zhang et al., 2020), model manipulations (Heo et al., 2019) and explanation-aware backdoors (Noppel et al., 2023) are also discussed in current literature. However, the present article focusses on privacy attacks that aim to compromise the personal data of individuals or the confidentiality of the underlying model. In the XAI context, model explanations further aid (Zhao et al., 2021) the identification or exposure of personal information of individuals or the intellectual property of the model owner. Table 2 provides an overview of these intentional privacy risks through XAI and the studies addressing them.

### 4.2.1 Membership inference

This is a privacy risk of identification of an individual in the training set of a model (Shokri et al., 2017; Zarifzadeh et al., 2024) (Figure 6). An adversary can execute this attack with black-box or white-box access to the model (Veale et al., 2018) after it has been deployed. A membership inference model can be expressed as the following binary classifier (Kuppa & Le-Khac, 2021) when the model output, $f(x)$, is available:

$$A_{MemInf} : x, f(x) \rightarrow \{member, non\text{-}member\} \tag{1}$$

When explanations, $\phi(x)$, are available, they can be alternatively used to differentiate members from non-members using the following inference model:

$$A_{MemInfExp} : x, \phi(x) \rightarrow \{member, non\text{-}member\} \tag{2}$$

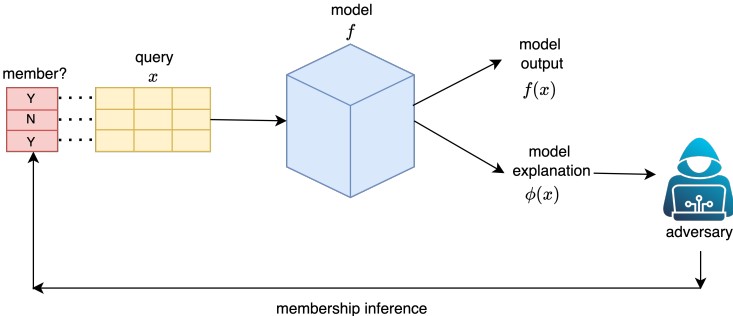

Figure 6: Membership inference exploiting explanations.

The seminal work on membership inference in feature-based and example-based XAI systems was presented by Shokri et al. (2020; 2021). The study used various backpropagation and perturbation methods to show the vulnerability of feature-based systems. The proposed attack used variances in the prediction and explanation vectors to differentiate between members and non-members. Liu et al. (2024) introduced a membership inference on feature-based XAI using model performance and robustness metrics. The study observed higher loss in confidence on perturbation of important features for members and utilized this observation in training an attack model, in addition to using the performance loss from the model. Ma et al. (2024) extended membership inference to label-only settings on Shapley value explanations. This method, which builds on earlier work on label-only attacks using hard prediction labels (Choquette-Choo et al., 2021), improved neighbourhood sampling using explanations thus reducing the number of queries.

In the example-based category, Shokri et al. (2020; 2021) investigated influence functions on logistic regression models. Since influence functions generate explanations in the form of actual datapoints, the study observed

that attackers could obtain certainty about membership, thus leading to stronger attacks. More recently, Cohen & Giryes (2024) considered self-influence functions instead, that show the influence of a datapoint on its own prediction. The proposed attack required white-box access to the target model parameters, activations, and gradients. The selection of an appropriate threshold range for self-influence scores associated with members was critical for this attack and was achieved by maximizing the balanced accuracy on the training set.

Kuppa & Le-Khac (2021) used a different type of example-based explanation, namely, counterfactuals, for membership inference. The authors trained shadow models using counterfactual samples and auxiliary datasets. A threshold on the difference in predictions of the attack and target models was used to determine membership. Pawelczyk et al. (2023) also targeted counterfactuals and proposed two types of attacks. The first relied on the distances between datapoints and their counterfactuals to differentiate between members and non-members. The second used a loss-based approach using a likelihood-ratio test (Carlini et al., 2022) that improved the attack.

Interpretable models using decision trees, and surrogate models created using Trepan algorithm (Craven & Shavlik, 1995), were evaluated for membership inference by Naretto et al. (2022). The study also examined the effect of overfitting on the attack. The success of membership inference was determined to be higher on both interpretable and surrogate models compared to black-box models. Further, surrogates of overfitted models exhibited higher susceptibiliy to the attack than those derived from well-regularized models.

Membership inference attacks in machine learning models have been explored extensively in existing literature (Hu et al., 2022) and attack strategies have exploited confidence scores and predictions (Shokri et al., 2017). However, the recent attacks that exploit explanations suggest that XAI interfaces provide a new avenue for adversaries to launch this attack. Such attacks have targeted feature-based, example-based, and interpretable (including surrogates) XAI methods. The effectiveness of the attack is influenced by factors such as dataset type (Shokri et al., 2021), dimension (Pawelczyk et al., 2023; Shokri et al., 2021), model architecture (Shokri et al., 2021) and overfitting (Pawelczyk et al., 2023). Some attacks have proven effective in the absence of knowledge of the training dataset or target architectures (Liu et al., 2024), underscoring their practical threat potential.

While interpretable models are often recommended as surrogates for explaining black-box models (McDermid et al., 2021), as demonstrated by these attacks, the layer of interpretability can introduce a backdoor to the target system and lead to privacy leaks (Naretto et al., 2022). In the example-based category, influence functions expose data instances, particularly outliers, due to their distinct characteristics and higher influence on the training process (Shokri et al., 2021). Among feature-based methods, those using perturbations exhibit higher resilience to membership inference due to use of out-of-distribution points, however, this can also result in reduced explanation fidelity ((Shokri et al., 2021). Conversely, feature-based methods with better explanation quality are also found to be susceptible to higher leakage (Liu et al., 2024) suggesting a conflict between privacy and utility.

### 4.2.2 Model inversion

This category of privacy risk can result in reconstruction of datapoints partly or completely from outputs (Zhang et al., 2023) (Figure 7). These attacks can be conducted with black-box or white-box access to the model (Fredrikson et al., 2015; Veale et al., 2018) after it has been deployed in production systems. Attribute inference is a type of data reconstruction that can determine the values of certain attributes, generally those sensitive to individuals (Yeom et al., 2018) from outputs. When the model output, $f(x)$, is used for inversion, the inference model can be expressed as follows:

$$A_{ModInv} : f(x) \to x \tag{3}$$

When explanations are available, they can be alternatively exploited to reconstruct datapoints using the following inference model:

$$A_{ModInvExp} : \phi(x) \to x \tag{4}$$

Model inversion attacks have been documented in XAI on example-based, feature-based, and interpretable systems. Shokri et al. (2020; 2021) demonstrated a data reconstruction attack on influence functions in

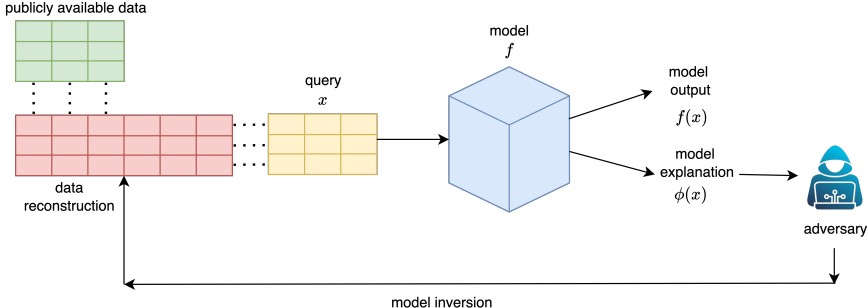

Figure 7: Model inversion exploiting explanations.

logistic regression models and found the attack dependent on data dimensionality. The authors designed different heuristics for low and high dimension data to improve coverage and efficiently recover more training points. Goethals et al. (2023) showed an explanation linkage attack using native counterfactuals generated from actual instances of the training data. The attack demonstrated the vulnerability of counterfactuals in leaking private attributes.

Private images were found to be susceptible to recovery through saliency maps by Zhao et al. (2021) leading to inadvertent exposure. The study found XAI systems that provided class-specific multiple explanations particularly prone to leakage. The authors also used attention transfer to highlight similar risks for non-explanation models. Other studies (Duddu & Boutet, 2022; Luo et al., 2022) have focused on attribute inference of tabular data using feature-based XAI. The former trained attack models using predictions and explanations to infer sensitive features. The latter used Shapley values and effectively executed the attack with limited number of queries on cloud ML services. Toma & Kikuchi (2024) further showed that the efficacy of the proposed attack was dependent on the combination of black-box architecture and XAI method. Their findings indicate that linear models using Shapley values were particularly vulnerable to attribute inference.

Ferry et al. (2024) designed a probabilistic white-box attack applicable to transparent models, such as decision trees and rule lists, and quantified the information about the training data contained in the model. The work found that models built using greedy algorithms leak more information compared to those built using optimal strategies. The authors also observed the attack's capacity for misuse in launching other inference attacks such as membership and property inference.

The above attacks from current literature demonstrate the potential of misuse of explanations APIs, resulting in a two-way flow (Veale et al., 2018) where inputs can be determined from outputs. Model explanations provide an effective attack surface compared to predictions (Duddu & Boutet, 2022; Zhao et al., 2021), indicating the contradiction between the need for explanations in Trustworthy AI and protecting privacy (Zhao et al., 2021). Data reconstruction attacks impact active users of AI systems, thus putting end-users at risk (Zhao et al., 2021) and having a higher impact. In certain proposed model inversion attacks, sensitive attributes can be retrieved from models trained on non-sensitive attributes (Duddu & Boutet, 2022) while other proposed attacks demonstrate higher leakage from more important attributes (Luo et al., 2022) and recovery of entire training datasets (Shokri et al., 2021). In addition, the above works highlight the misuse of XAI techniques even for models that do not provide explanations (Zhao et al., 2021).

A tension is also found between preserving privacy and maintaining utility of the XAI system. For instance, synthetic counterfactuals created by perturbing actual samples are shown to provide resilience to inversion in comparison to native counterfactuals. However, their usage is found to affect the plausibility and runtime of explanations (Goethals et al., 2023), suggesting degrading utility. Similarly, the use of multiple diverse explanations are usually recommended (Vo et al., 2023) for improving understandibility of explanations, however they are also found to contribute to leakage of privacy. Consequently restricting the access to explanation APIs has been suggested as a countermeasure (Zhao et al., 2021), however such restrictions may reduce the utility to end-users.

### 4.2.3 Model extraction

In this attack, the target model is replicated to a significant degree of accuracy and fidelity (Jegorova et al., 2022), thus breaching the confidentiality and the intellectual property of the model owner (Figure 8). In a typical model extraction attack, the adversary has black-box access to a deployed victim model and uses an unlabeled dataset to query it, thus building an attack dataset (Yan et al., 2023b) for cloning the model. In contrast, data-free model extraction leverages generative models to synthesize the datasets, which is advantageous when the input data is difficult to obtain (Miura et al., 2024). When the model output, $f(x)$, is utilized for this attack, the extraction of the target model function, $f'$, can be expressed as follows:

$$A_{ModExt} : x, f(x) \rightarrow f' \tag{5}$$

When explanations are utilised for the purpose of extraction, the attack model can be expressed as follows:

$$A_{ModExtExp} : x, \phi(x) \rightarrow f' \tag{6}$$

Since the extracted models can further leak personal data through membership inference and model inversion (Song et al., 2017), this attack is usually used as a starting point for initiating other types of attacks (Aïvodji et al., 2020; Miura et al., 2024).

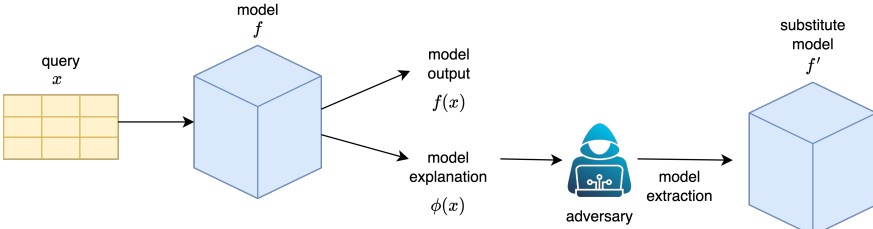

Figure 8: Model extraction exploiting explanations.

Model extraction attacks using explanations have been proposed across feature-based and example-based XAI methods. In the seminal work on the topic, gradient explanations, in the form of saliency maps, were found to be vulnerable by Milli et al. (2019). The use of explanations improved the attack by reducing the number of queries compared to attacks relying solely on model predictions. Miura et al. (2024) also leveraged gradient-based explanations but used data-free approach to train generative models for creating the attack dataset. The inclusion of explanations improved the quality of the generative model, and the accuracy of the cloned model improved with the diversity of the generated samples. Similarly, Yan et al. (2023a) employed data-free extraction wherein explanation loss was used to guide the generative model. Accuracy of the cloned model was improved by matching the victim model's predictions and explanations.

A different approach of extraction on feature-based XAI, used multitask learning to learn both classification and explanation tasks of the victim model (Yan et al., 2022). Further, a model agnostic technique on gradient and pertubation-based XAI (Yan et al., 2023b), showed that explanations provided auxiliary information that enabled more efficient attacks, reducing the query budget compared to prediction-only strategies. The attack could also be applied to non-explanation models and achieved accuracy equivalent to those of explanation models.

Besides the above extraction attacks targeting various feature-based XAI, from the example-based category, counterfactuals have been mainly targeted for this attack. In an extraction attack proposed by Aïvodji et al. (2020), they were used to approximate the decision boundary of the victim model with high accuracy and fidelity under low query budgets. Multiple and diverse counterfactuals were found to aid the extraction process by divulging additional information to adversaries. An improvisation of the attack, to reduce the number of queries further, mitigate decision boundary shift and achieve higher agreement with the victim model, was proposed by Wang et al. (2022). The method used the original counterfactual explanation with its own counterfactual as training pairs, to extract additional datapoints to train the cloned model. In

another approach, Kuppa & Le-Khac (2021) proposed iterative querying of the victim model to capture the training data distribution. The method utilized distillation loss to transfer knowledge from the victim to the cloned model and was found to be successful due to the optimization of various properties such as diversity, proximity, feasibility, and sparsity.

As demonstrated by the aforementioned attacks, explanation-based extraction attacks offer substantial advantages over traditional prediction-only approaches by facilitating model replication with reduced number of queries (Milli et al., 2019; Miura et al., 2024). The reduction in the number of queries benefits the adversary, especially in pay-by-query models. Certain attacks are also possible with partial knowledge of the data distribution (Aïvodji et al., 2020) or in absence of overlap between the attack and training datasets (Yan et al., 2022). In addition, in scenarios where attackers do not possess the input datasets, data-free extraction attacks are possible and the use of explanations is shown to improve the attack accuracy (Miura et al., 2024). Moreover, the diversity of the generated input datasets in such attacks is found to improve the accuracy of the cloned models (Miura et al., 2024).

In addition to the direct threat to explanation models, XAI techniques can be misused for extraction of non-explanation models (Yan et al., 2023b). The use of diverse explanations, intended to build user trust in explanations, can lead to further leakage of privacy (Aïvodji et al., 2020). Similarly, the optimization of counterfactuals to satisfy various properties to improve explanation quality, can reveal information to adversaries about class-specific decision boundaries thus aiding the attack (Kuppa & Le-Khac, 2021) and leading to the conflict of explainability and privacy with utility.

### 4.2.4 Property inference

This type of risk pertains to inference of properties from the training data such as global statistics or aggregates (Mahloujifar et al., 2022), which model owners did not intend on sharing (Ganju et al., 2018) (Figure 9). The inferred property may not correspond to features in the training data or be correlated to any feature. A property inference model that uses the model outputs, $f(x)$, to infer such property, $p$, can be expressed as the following:

$$A_{PropInf} : f(x) \to p \tag{7}$$

When explanations, $\phi(x)$, are available, the inference model can be expressed as follows:

$$A_{PropInfExp} : \phi(x) \to p \tag{8}$$

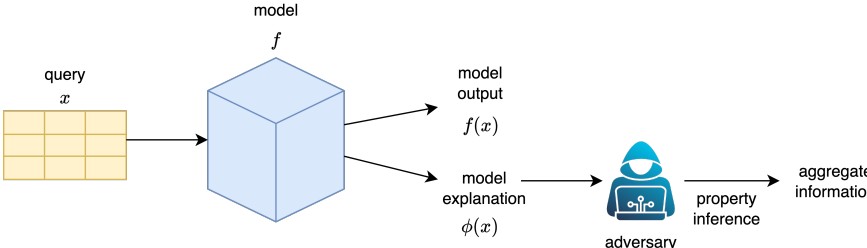

Figure 9: Property inference exploiting explanations.

Although property inference is a known issue in AI models, to the best of our knowledge, no attacks have yet been documented that exploit explanations for this purpose.

**Table 2** Studies on intentional privacy leakage in XAI systems.

| Privacy Risk | XAI Category | XAI Method | Study |
|---|---|---|---|
| Membership inference | Interpretable | Decision tree | Naretto et al. (2022) |
| | Interpretable (surrogate) | Trepan | Naretto et al. (2022) |
| | Example-based | Influence functions | Shokri et al. (2020; 2021) |
| | | Counterfactuals | Kuppa & Le-Khac (2021); Pawelczyk et al. (2023) |
| | | Self-influence functions | Cohen & Giryes (2024) |
| | Feature-based | Gradient, integrated gradients, guided backpropagation, LRP, LIME, SmoothGrad | Shokri et al. (2020; 2021) |
| | | Integrated gradients, SmoothGrad, VarGrad, Grad-CAM++, SHAP, LIME | Liu et al. (2024) |
| | | Shapley values | Ma et al. (2024) |
| Model inversion | Interpretable | Decision tree, rule list | Ferry et al. (2024) |
| | Example-based | Influence functions | Shokri et al. (2020; 2021) |
| | | Counterfactuals (native) | (Goethals et al., 2023) |
| | Feature-based | Gradient, gradient x input, class activation maps (CAM), Grad-CAM, LRP | (Zhao et al., 2021) |
| | | Integrated gradients, DeepLIFT, GradientSHAP, SmoothGrad | Duddu & Boutet (2022) |
| | | Shapley values | Luo et al. (2022); Toma & Kikuchi (2024) |
| Model extraction | Example-based | Counterfactuals | Aïvodji et al. (2020); Kuppa & Le-Khac (2021); Wang et al. (2022) |
| | Feature-based | Gradient | Milli et al. (2019) |
| | | Gradient, Grad-CAM, MASK | Yan et al. (2022) |
| | | Gradient, Grad-CAM, MASK, LIME | Yan et al. (2023b) |
| | | Grad-CAM, LIME | Yan et al. (2023a) |
| | | Gradient, integrated gradients, SmoothGrad | (Miura et al., 2024) |
| Property inference | Not reported | Not reported | - |

### 4.3 Unintentional privacy leakage

This subsection discusses unintentional privacy leakage in XAI that occur without malicious intent (Jegorova et al., 2022). Certain leakages can occur due to the natural mechanisms of the training process or through the content of explanations. These may occur during the different AI lifecycle phases or during the course of an explanation (Rawal et al., 2022).

### 4.3.1   Training issues

Training issues such as, overfitting and memorization, identified in AI models can lead to privacy leakage. Overfitting is found to aid membership and attribute inference attacks (Yeom et al., 2018). Memorization leads to the model remembering subsets of training data (Song et al., 2017) and occurs during training before overfitting begins (Jegorova et al., 2022). It can cause leakage when data owners deploy models with codebases and training pipelines developed by third parties, such as in MLaaS, allowing sensitive information to be leaked from training data (Song et al., 2017).

### 4.3.2   Explanation content

The content of explanations may contain values of sensitive fields. For instance, in example-based explanations such as influence functions, training datapoints potentially containing sensitive fields, are directly revealed to end-users. Karimi et al. (2023) provide another example of unintentional leakage through contrastive explanations, which can lead to inference of sensitive details of individuals whose partial attributes are known. Additionally, interpretable models used as surrogates, can reveal properties of the training data or additional information about the black-box beyond what the model owner intended to share (Blanco-Justicia et al., 2020). In addition to direct content-based leakage, risks may also arise from the inadvertent exposure of explanations to unauthorised users (Kuppa & Le-Khac, 2021). For example, during troubleshooting of error cases, developers or quality engineers may unintentionally access sensitive personal information in the explanation. Moreover, even when direct identifiers are absent, explanations that contain proxy or correlated features can still enable indirect inference.

## 5   Privacy Preservation Methods in XAI

To address the privacy risks outlined in Section 4, a growing body of research has emphasized the need for privacy preserving XAI techniques (Aïvodji et al., 2020; Shokri et al., 2021; Zhao et al., 2021). In response to these concerns, several studies have proposed methods to generate explanations while mitigating privacy concerns. Many of these approaches draw upon established principles and methods from the broader domain of privacy preserving ML (PPML), adapting them to specific challenges posed by explanation techniques. This section synthesizes the key contributions in literature to enhance the privacy guarantees of XAI systems in alignment with the objectives of RQ2. We categorize the proposed solutions under the main approaches in PPML. Table 3 summarizes these approaches and methods, and Figure 10 presents a consolidated mapping of privacy preserving strategies to specific types of privacy attacks discussed earlier.

### 5.1   Differential privacy

Differential privacy (DP) (Dwork et al., 2006) is a widely recognised method that provides a quantifiable definition of privacy and the incremental privacy loss from publishing confidential data (McKay Bowen & Garfinkel, 2021). A mechanism is differentially private if it can hide the participation of any single individual in a dataset (Harder et al., 2020). This can be achieved by using noise and is typically associated with an adverse effect on the accuracy of the system (Harder et al., 2020). By adjusting the privacy budget, $\epsilon$, from 0 to $\infty$, practitioners can manage this trade-off between privacy and accuracy (McKay Bowen & Garfinkel, 2021). Given its robust privacy guarantees, early research in privacy preserving explanations has adopted DP using various strategies to safeguard the training data in interpretable, feature-based, and example-based XAI. In the context of XAI, an explanation is differentially private if it can obscure any single individual in the training dataset (Patel et al., 2022). The technique can be applied at various stages, including the explanation generation algorithm (Patel et al., 2022), the training process of the target model (Cohen & Giryes, 2024; Liu et al., 2024) or directly on the training data (Bozorgpanah & Torra, 2024; Ezzeddine et al., 2024).

Decision trees are popular due to their simplicity and inherent interpretable qualities, however they are prone to privacy leakage (Ferry et al., 2024; Naretto et al., 2022). A number of algorithms have been developed for building decision trees based on DP guarantees (Fletcher & Islam, 2020) offering different trade-offs in privacy and utility. The interpretability of private trees and their subsequent usefulness to XAI users,

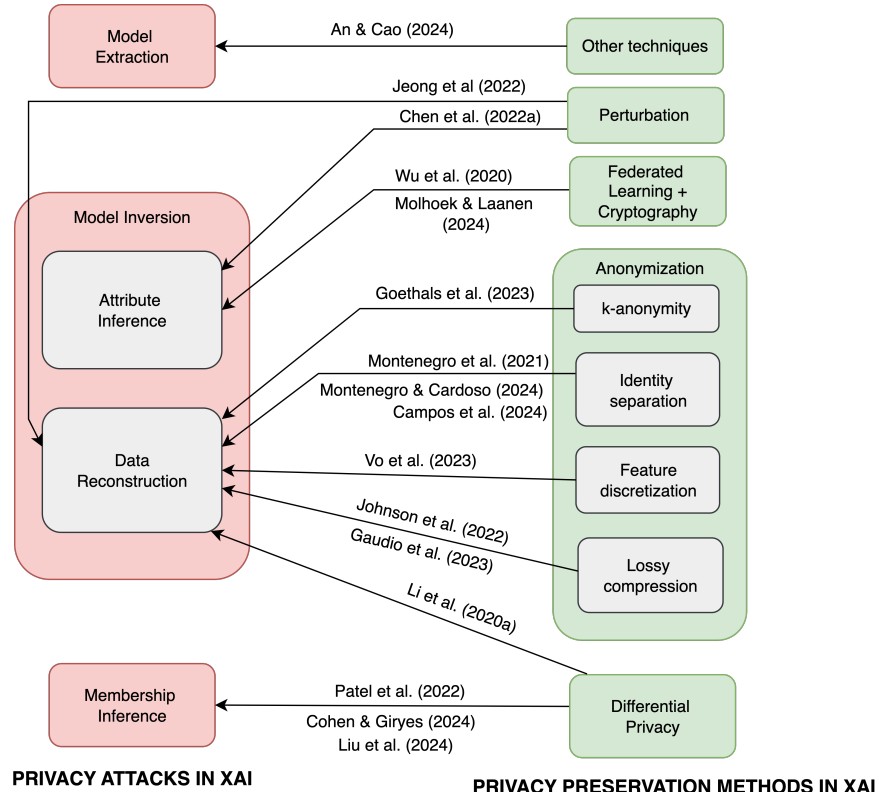

Figure 10: Proposed privacy preservation methods for specific privacy attacks in XAI.

depend on factors such as the privacy budget per query, tree depth, pruning strategies and termination criteria (Fletcher & Islam, 2020). Nori et al. (2021) applied DP to another type of interpretable model, namely, Explainable Boosting Machines (EBM), to prevent privacy leakage of training data. The resulting privatized system demonstrated good accuracy at low privacy budgets while facilitating correction of errors introduced by noise, the removal of bias and the enforcement of constraints such as monotonicity. Building on this, Baek & Chung (2024) further optimized the utilization of privacy budget in these models to improve accuracy, incorporating techniques such as gradient error optimization and pruning of non-essential features.

Harder et al. (2020) proposed an interpretable model using differentially private locally linear maps with Gaussian mechanism per output class. The filters learned by the model from input images were observed to have higher interpretability compared to feature-based methods, such as integrated gradients and Smooth-Grad. However, increasing the number of such maps per class dropped accuracy due to the distribution of privacy budget over additional parameters. In a different approach, Li et al. (2020a) proposed an interpretable model in the form of feedforward-designed convolutional neural network (FF-CNN) made privacy preserving by using DP guarantees on subspace approximation with adjusted bias (Saab). Their findings indicated the the integration of DP was effective in mitigating the risk of reconstruction of input images while maintaining classification accuracy.

For feature-based explanations generated by local linear approximations around the point of interest, Patel et al. (2022) introduced a differentially private approach for loss calculation in the explanation algorithm. The study also proposed an adaptive method of reusing previous explanations for prudent usage of the privacy budget. Nguyen et al. (2023) employed local DP to restrict adversaries from learning the top influential features through aggregated scores in feature-based XAI. While originally proposed to defend against a backdoor security attack exploiting explanations (Severi et al., 2021), the random perturbation of influential features under local DP guarantees was observed to preserve the privacy of those features while maintaining explanation fidelity. Bozorgpanah & Torra (2024) also applied local DP to mask the training

dataset and investigated its impact on privacy and utility of feature-based explanations. They introduced an irregularity metric to measure the feature distortion due to privatization of the original dataset and the change in explanation values. Their findings indicate that the use of additive noise on the training dataset caused irregularities, thereby reducing the utility of the explanations. In a related study, Ezzeddine et al. (2024) added calibrated noise to training datasets and evaluated the impact on SHAP explanations using various distance metrics. They observed the change in SHAP values in the privatized systems correlated with the privacy budget and data dependent. Abbasi et al. (2024) used a different approach on the data and employed DP for synthetic data generation for training of different model architectures. They used similarity scores to track the change in explanations while utility loss evaluated the drop in accuracy, thus quantifying the triad of privacy, utility and explainability.

In addition to the aforementioned studies, researchers have also examined the use of DP in the training process of the target model in feature-based (Liu et al., 2024) and example-based (Cohen & Giryes, 2024; Mochaourab et al., 2023) XAI. These investigations have determined mitigation against membership inference attacks for high privacy budgets (Cohen & Giryes, 2024; Liu et al., 2024). The introduction of DP noise serves as a regularization mechanism for target models (Nori et al., 2021) and its mathematical guarantee enables quantification of privacy, making it a compelling choice as a privacy enhancing technology. In the context of XAI, in addition to its application in mitigating membership inference from explanations (Cohen & Giryes, 2024; Liu et al., 2024; Patel et al., 2022), it is also found to mitigate reconstruction of sensitive inputs (Li et al., 2020a). However, the improvement in mitigation of attacks at high privacy budgets and hence degrading accuracy (Cohen & Giryes, 2024; Liu et al., 2024) can be a setback to the use of this technique. In addition to adversely affecting accuracy, its introduction can also deteriorate explanation quality in terms of fidelity (Liu et al., 2024; Patel et al., 2022) with pronounced effects on minority groups (Patel et al., 2022). The technique is also ineffective against attribute inference attacks when there are existing strong correlations between different attributes (Chen et al., 2022a). In such cases, the privacy enhancing benefits of DP may be insufficient to prevent adversaries from inferring sensitive attributes.

Algorithms such as DPSaab (Li et al., 2020a), have been observed to offer a more favourable trade-off between accuracy and privacy. Practitioners can employ strategies such as reusing previously generated differentially private explanations to utilize the privacy budget effectively (Patel et al., 2022). Methods that use local DP, are observed to achieve high faithfulness of explanations with privacy (Nguyen et al., 2023), thus demonstrating that it is possible to balance multiple desirable properties. Hence judicious use of this technique can ensure that privacy is achieved while maintaining reasonable utility of the model and explanations.

## 5.2 Cryptography

Cryptographic protocols for privacy preservation in ML use secure algorithms to protect the target model and data. Prominent methods in this category include homomorphic encryption, secret sharing, and secure multi-party computation (Yin et al., 2022). In XAI, these techniques have seen limited application in interpretable and example-based systems. They have also been used in conjunction with other privacy preserving techniques such as federated learning (El Zein et al., 2024; Molhoek & Laanen, 2024; Wu et al., 2020; 2023).

For interpretable tree-based models, Zhao et al. (2023) proposed an additive homomorphic scheme for model owners and query users, for pushing the encrypted model and query data to cloud service providers for inferencing. Adding perturbations to the inference results and query data ensured privacy protection of these assets while maintaining accuracy comparable to non-private inference. In the example-based category, Veugen et al. (2022) proposed a cryptographic method with secure multi-party computation to generate contrastive explanations, while protecting private training data and confidentiality of the model. The algorithm securely trained a binary decision tree to generate fact and foil leaves, which were used as explanations for a query datapoint. Additionally, a synthetic datapoint from the foil leaf was provided to the end-user to enhance explainability.

Cryptographic methods, such as homomorphic encryption, introduce significant computational complexity in the system (Liu et al., 2022a). The use of encryption can deteriorate model transparency, limiting the

ability of data scientists to correct errors, inspect data, add features or fine tune the model (Dowlin et al., 2016). Therefore it is essential to implement cryptographic protocols in XAI system components in the right use cases to complement other privacy preservation techniques or when other techniques are infeasible or costly.

## 5.3 Anonymization

Anonymization refers to the process of transformation of data (Majeed & Lee, 2021) to obscure the distinctive features of individuals, thus safeguarding their privacy. The process is associated with the removal or modification of direct and quasi-identifiers (Majeed & Lee, 2021), that can uniquely identify individuals. Various methods of anonymization are used in practice, such as k-anonymity, l-diversity, and t-closeness (Yin et al., 2022). In XAI, different anonymization techniques have been demonstrated in example-based, feature-based, and interpretable methods. Techniques such as disentangled representation learning and lossy compression have been applied on sensitive visual data, such as medical images, to generate privatized explainable-by-design representations.

A dataset is considered k-anonymous if every record is indistinguishable from k-1 other records (Sweeney, 2002b), thus providing a measure of the risk of re-identification of records. K-anonymity can be achieved using methods such as suppression and generalization (Sweeney, 2002a), which obscure the data to remove identifiable features. Though traditionally this technique is applied to target datasets for protection, Goethals et al. (2023) proposed its usage on native counterfactuals, that are actual datapoints from the training dataset, for protection against model inversion attack through explanations. This strategy of generating k-anonymous counterfactuals was shown to result in lower information loss and higher validity, outperforming counterfactuals derived from k-anonymized datasets. Further analysis by Berning et al. (2024) determined that the effectiveness of k-anonymous counterfactuals is confined to dense areas of the dataset. Its offered privacy protection was also found disproportionate to the value of k. Vo et al. (2023) highlighted another limitation, namely, the computational overhead of generating these counterfactuals requiring querying the explainer for a large number of counterfactuals. The authors proposed an alternative strategy of privatizing diverse counterfactuals by discretization of continuous features. This technique is closely related to generalization in privacy preserving data mining and is particularly effective against linkage attacks.

K-anonymity using microaggregation has been applied on the training dataset in feature-based XAI by Bozorgpanah & Torra (2024). The study found explanations from non-private and private datasets largely aligned, with minor irregularities observed. The alignment indicated that utility was effectively preserved after privatization. Similarly, Blanco-Justicia et al. (2020) applied microaggregation to build local tree-based surrogate explanations from clusters around an instance to be explained. The method enforced k-anonymity by restricting the cluster size and incorporated shallow trees to enable comprehensibility.

An emerging area of study focusses on providing explanations while protecting privacy in the medical domain, where privacy of patients' visual data is crucial. The primary aim of such techniques is transformation of private data through removal of identifying features while retaining explanatory evidence. Strategies such as the use of autoencoders for disentanglement of identifiable characteristics (Montenegro & Cardoso, 2024), Siamese network for increasing identity distance between original and privatized images (Montenegro et al., 2021) and latent diffusion models for generating synthetic images Campos et al. (2024) are proposed. The use of lossy compression by pixel sampling is also observed to remove identification information while being post-hoc explainable (Gaudio et al., 2023; Johnson et al., 2022). This approach also has an added advantage of reducing the image size significantly, thus making medical training datasets smaller (Gaudio et al., 2023).

In critical domains, such as healthcare, anonymizing training and query data can assist in protecting identifiable information. However, the applied techniques should preserve the output quality for utility to diverse end-users (Campos et al., 2024). Unlike DP, anonymization techniques lack proven guarantees, however despite DP's theoretical guarantees, it is unable to scale beyond low resolution image data (Campos et al., 2024). Lossy compression alternatively provides an effective way of privatizing image data, with the benefits of achieving both privacy and explainability while reducing training dataset sizes. It thus enables data sharing with multiple parties in non-private settings (Gaudio et al., 2023) and can serve as an explanation generation method for sensitive image data.

Anonymization techniques, such as k-anonymity, protect privacy of individuals by mitigating re-identification and linkage attacks (Vo et al., 2023). When applying k-anonymity, selecting an appropriate value of k is critical in striking the right balance between accuracy and privacy risk level (Bozorgpanah & Torra, 2024). While higher values of k enhance privacy, explainability may be adversely affected (Berning et al., 2024; Blanco-Justicia et al., 2020). Moreover, the actual level of privacy may also not scale with increasing values of k (Berning et al., 2024). Hence the selection of k that achieves the right trade-off in privacy, explainability and utility is important. Additionally, k-anonymity has limitations such as its dependence on data characteristics, susceptibility to homogeneity attack (Berning et al., 2024), and its vulnerability to privacy leakage when background knowledge is available or diversity is lacking in the private attributes (Goethals et al., 2023). Other techniques such as l-diversity and t-closeness may address some of these challenges, though their applicability to explanations remain unexplored.

The generation of synthetic data for privacy preserving data analysis is explored in previous non-XAI works (Boedihardjo et al., 2023; Liu et al., 2022b). Generating synthetic data that is private, accurate and preserves properties of the true data is a known challenge and NP-hard in the worst case (Ullman & Vadhan, 2011). When models are trained on such data, the explanations through XAI tools are expected to be inherently privacy preserving, hence this approach can be a promising direction in preserving privacy in explainable systems.

## 5.4   Perturbation

Perturbation of sensitive data is a widely recognized technique in the field of privacy preserving data publishing (Tran et al., 2024; Yin et al., 2022). When explanations contain sensitive information, obfuscating the contents through perturbations can prevent direct exposure. This technique can also be applied to stem indirect leakage of inferencing sensitive attributes through explanations.

Chen et al. (2022a) proposed a generic privacy preserving mechanism applicable to different XAI types such as feature-based and interpretable surrogates. The proposed method perturbed the decision mapping of an algorithm prior to public release of transparency reports. To mitigate privacy leakage while upholding utility, the authors defined a maximum confidence measure in the inference of sensitive attributes of data subjects and a utility measure in terms of faithfulness. Jeong et al. (2022) applied perturbations on saliency map explanations as a defense mechanism for model inversion in image models. The proposed framework comprised of a two-player minimax game between inversion and noise injector networks, in which the inversion network attempted to reconstruct images from saliency maps and the noise injector perturbed explanations to counter the inversion. The use of multiple evaluation metrics to differentiate between original and reconstructed images facilitated the quantification of the privacy of the defense mechanism.

For the prevention of privacy leakage in XAI, researchers have attempted perturbation of two types of model outputs, namely, predictions and explanations. Adding perturbations to model predictions, such as the strategy of adding noise to output confidence scores used by MemGuard (Jia et al., 2019), is found ineffective in mitigating membership inference through explanations (Liu et al., 2024). Perturbation of explanations is also insufficient in defending against data-free model extraction based on explanations (Yan et al., 2023a). However, the strategy has shown promising results in countering model inversion. The use of perturbation techniques at the explanation interface is also attractive due to its ease of implementation that requires no retraining of the model (Jeong et al., 2022). Nevertheless, large magnitude noise can degrade the usefulness of explanations (Jeong et al., 2022), hence perturbations should be carefully calibrated to minimize any adverse impact on explanation quality.

## 5.5   Federated learning

Among the distributed privacy enhancing techniques available in PPML, Federated learning (FL) is an architectural solution (El Mestari et al., 2024) that enables training of local models on user devices and exchange of model parameters with a centralized server that co-ordinates the training of a shared global model (Konečný et al., 2016). It thus enables collaborative learning while keeping users' private data at the source (Guerra-Manzanares et al., 2023) and mitigates the privacy risk of multiple parties sharing their sensitive data with other parties (El Zein et al., 2024) or a centralized server (Zhu et al., 2022). In horizontal

federated learning (HFL), local datasets have the same feature space but contain different samples while vertical federated learning (VFL), involves datasets with different feature spaces but overlaps in samples (Fiosina, 2022).

To address both privacy and explainability in Trustworthy AI, the combination of FL and XAI, i.e., Fed-XAI is suggested (Bárcena et al., 2022; Corcuera Bárcena et al., 2023; López-Blanco et al., 2023) and refers to the federated learning of XAI models. Many approaches of Fed-XAI using HFL and VFL are proposed in literature. Fiosina (2022) used a HFL approach for forecasting taxi trip duration and applied feature-based explainability methods post-hoc. Chen et al. (2022b) used an explainable VFL framework to optimize counterfactual explanations using a representative query distributed on multiple parties. Both setups demonstrate the use of post-hoc explainability tools in a distributed environment, with FL serving as a privacy preserving setup for collaborative learning of sensitive data owned by multiple parties. Fed-XAI architectures have also leveraged interpretable models locally, such as fuzzy rule-based classifiers (Daole et al., 2024), Takagi-Sugeno (Zhu et al., 2022) and Takagi-Sugeno-Kang (Corcuera Bárcena et al., 2023) fuzzy rule-based models. In these setups, interpretability is achieved using underlying explainable-by-design (Corcuera Bárcena et al., 2023) models.

Though FL aids privacy by default, it is prone to reconstruction and inferencing attacks (Mothukuri et al., 2021; Zhang et al., 2024). The sharing of gradients and model parameters, communication mechanism and aggregation process can lead to leakage of privacy of the participating clients (Zhang et al., 2024). Hence researchers have proposed integration of other privacy preserving techniques, such as cryptography, with FL methods. In one such work, Molhoek & Laanen (2024) generated synthetic data on vertically partitioned data in a FL two-party setup. Counterfactuals built from this synthetic data using secure multi-party computation, were ranked and shared with both parties and were found to be resilient to attribute inference. El Zein et al. (2024) proposed a HFL structure using decision tree models, wherein a global decision tree was collaboratively trained by participants and additive secret-sharing was used in aggregation of intermediate results. A VFL technique, Falcon (Wu et al., 2023), utilized a hybrid approach consisting of partially homomorphic encryption (PHE) and additive secret sharing for exchange of intermediate computations. Another setup, Pivot (Wu et al., 2020), proposed as part of Falcon, used threshold partially homomorphic encryption (TPHE) and additive secret sharing to protect privacy of intermediate exchanges. Though these works successfully integrate cryptographic techniques with FL, research has also determined that the use of cryptographic methods in FL reduces the centralized server's ability to differentiate true model parameters leading to backdoor attacks (Guo et al., 2022). Hence appropriate defense frameworks, such as trust evaluation schemes (Guo et al., 2023), should be incorporated for protection of the FL system from malicious users.

FL enables the training of AI models from diverse, private, and high-quality data (Zhu et al., 2022) located at client systems. It reduces the footprint of user data in the network (Mothukuri et al., 2021) by keeping data at the source and avoids transmission and storage of sensitive information in a centralized location when multiple parties are involved (Wang et al., 2019). Despite its benefits, in its current form FL faces challenges for its risk-free adoption (Mothukuri et al., 2021) including ensuring privacy constraints, merging of local XAI models and dealing with large data streaming that can lead to concept drifts (Bárcena et al., 2022). The introduction of XAI methods in the FL architecture, can also further increase the vulnerability of the system to privacy attacks through explanations. Thus Fed-XAI presently cannot guarantee privacy preservation through XAI components and further research to develop strategies to stem inadvertent privacy leakage through explanations is essential.

## 5.6 Other techniques

In addition to the main privacy preservation methods commonly employed in PPML, certain non-standard techniques have also been explored to mitigate privacy leakage in certain types of XAI. These approaches aim to enhance privacy preservation by adopting alternative strategies including limiting access to training data, obscuring features, or providing an abstraction of the target models. While not traditionally classified under formal privacy methods, these approaches contribute to reducing privacy leakage and complement other methods.

A client-centric, data-driven approach of generating counterfactuals was proposed by An & Cao (2024) by leveraging previous inferences retrieved by the model user. Due to the generation of counterfactuals locally at the client, the method was shown to be resilient to model extraction while achieving desirable properties such as diversity and succinctness. In another approach to create an interpretable model from a neural network, Marton et al. (2024) described a data-free strategy of distilling the function represented by the model. The method used synthetic data to train a set of neural networks and extracted the parameters to train an Interpretation-Net with an output representation in the form of surrogate decision trees.

Using a knowledge-based approach, Rožanec et al. (2022) applied semantic technologies in the form of domain specific ontology and knowledge graphs, to enhance explanations and describe features on a higher conceptual level. This enabled delinking explanations from features, thus preserving the confidentiality of the underlying model. Further, the integration of feature-based XAI such as LIME, enabled the system to determine features important for predictions. Terziyan & Vitko (2022) also applied semantic techniques to build XAI consisting of decision trees and rules generated from targeted points around the decision boundary of black-box models, without accessing the original training data. Due to the interoperability of semantic rules, the method enabled usage in decentralized setup for collaborative decision making, without individual parties sharing private local data.

These works demonstrate the application of data-free and knowledge-driven techniques in XAI to build privacy-by-design systems for protection of training data and the confidentiality of the model. By disconnecting features from the model and creating abstraction layers for generation of explanations (Rožanec et al., 2022), these strategies are helpful in protecting the underlying assets.

**Table 3** Privacy preserving methods applied to XAI systems.

| Privacy Preservation Category | Privacy Preserving Algorithm | Protected Asset | XAI Category (Method) | Study |
|---|---|---|---|---|
| Differential privacy | Various DP training algorithms | Training data | Interpretable (decision trees) | Fletcher & Islam (2020) |
| | DP locally linear maps | Training data | Interpretable (locally linear maps) | Harder et al. (2020) |
| | DPSaab | Training data | Interpretable (FF-CNN) | Li et al. (2020a) |
| | DP-EBM | Training data | Interpretable (EBM) | Baek & Chung (2024); Nori et al. (2021) |
| | DP explanation generation | Training and query data | Feature-based methods using local linear approximations (LIME, etc.) | Patel et al. (2022) |
| | Local DP | Training data | Feature-based methods that aggregate scores (SHAP, etc.) | Nguyen et al. (2023) |
| | DP trained SVM | Training data | Example-based (counterfactuals) | Mochaourab et al. (2023) |
| | DP-SGD | Training data | Feature-based (Grad-CAM) | Liu et al. (2024) |
| | DP-RMSProp | Training data | Example-based (self-influence functions) | Cohen & Giryes (2024) |
| | Local DP | Training data | Feature-based (TreeSHAP) | Bozorgpanah & Torra (2024) |

| Privacy Preservation Category | Privacy Preserving Algorithm | Protected Asset | XAI Category (Method) | Study |
|---|---|---|---|---|
| | Local DP | Training data | Feature-based (SHAP) | Ezzeddine et al. (2024) |
| | DP-WGAN (Wasserstein GAN) | Training data | Various XAI methods from DALEX framework[1] | Abbasi et al. (2024) |
| Cryptography | Privacy preserving foil trees | Training data, model | Example-based (contrastive explanations) | Veugen et al. (2022) |
| | Additive homomorphic encryption | Query data, inference results, model | Interpretable (tree-based models) | Zhao et al. (2023) |
| Anonymization | Microaggregation (MDAV) | Training data, model | Interpretable (decision trees) | Blanco-Justicia et al. (2020) |
| | Privacy preserving generative model | Training data | Example-based (case-based) | Montenegro et al. (2021) |
| | HeartSpot (lossy compression) | Training data | Feature-based (saliency maps) | Johnson et al. (2022) |
| | Discretization of features (generalization) | Training data | Example-based (counterfactuals) | Vo et al. (2023) |
| | CF-K (k-anonymity of counterfactuals) | Training data | Example-based (native counterfactuals) | Berning et al. (2024); Goethals et al. (2023) |
| | DeepFixCX (lossy compression) | Training data | Feature-based (saliency maps) | Gaudio et al. (2023) |
| | Microaggregation (MDAV) | Training data | Feature-based (TreeSHAP) | Bozorgpanah & Torra (2024) |
| | Disentangled representation learning | Training data | Example-based (case-based) | Montenegro & Cardoso (2024) |
| | Latent diffusion models | Training data | Example-based (case-based) | Campos et al. (2024) |
| Perturbation | GNIME | Training and query data | Feature-based (saliency maps) | Jeong et al. (2022) |
| | Linear-Time Optimal Privacy Scheme | Training and query data | Various XAI methods (interpretable surrogates, feature-based, etc.) | Chen et al. (2022a) |
| Federated learning | Pivot (VFL, additive secret sharing, TPHE) | Training data | Tree-based models (transparent) | Wu et al. (2020) |

---

[1]DALEX framework is available on `https://github.com/modeloriented/dalex`

| Privacy Preservation Category | Privacy Preserving Algorithm | Protected Asset | XAI Category (Method) | Study |
|---|---|---|---|---|
| | HFL | Training data | Feature-based methods (DeepLIFT, integrated gradients, LIME, etc.) | Fiosina (2022) |
| | HFL | Training data | Interpretable (Takagi-Sugeno,Takagi–Sugeno–Kang, fuzzy rule-based classifier) | Corcuera Bárcena et al. (2023); Daole et al. (2024); Zhu et al. (2022) |
| | VFL | Training data | Counterfactuals | Chen et al. (2022b) |
| | Falcon (VFL, additive secret sharing, PHE) | Explanations and training data | Feature-based (LIME) | Wu et al. (2023) |
| | PrivaTree (HFL, additive secret sharing) | Training data | Decision trees (transparent) | El Zein et al. (2024) |
| | VFL, SMC, Synthetic data | Query data | Example-based (counterfactuals) | Molhoek & Laanen (2024) |
| Other techniques | Semantic XAI | Training data | Interpretable (decision trees, semantic rules) | Terziyan & Vitko (2022) |
| | Semantic technologies (knowledge graphs, ontologies) | Model | Feature-based (LIME, etc.) | Rožanec et al. (2022) |
| | Guarded counterfactuals | Training data, model | Example-based (counterfactuals) | An & Cao (2024) |
| | Interpretation-Nets | Training data | Interpretable (decision trees) | Marton et al. (2024) |

## 6   Privacy Preserving XAI Characteristics

In the preceding sections, we have examined the privacy risks in XAI arising from both intentional and unintentional causes. We have also reviewed applicable privacy preserving methods to safeguard the additional attack surface exposed by explanations. In this section, drawing on the insights from investigation into RQ1 and RQ2, we aim to address RQ3 by identifying key characteristics that XAI should possess to mitigate the identified risks. These characteristics provide a framework for understanding the essential properties of privacy preserving XAI, taking into account the vulnerable assets that require protection and the various stakeholders involved during the AI lifecycle. The proposed characteristics offer guidelines to both researchers and practitioners to assess the effectiveness of existing privacy preserving XAI methods and guide the development of new approaches that prioritize privacy by design. By incorporating these qualities, XAI can strive to achieve the optimal balance between the triad of privacy, explainability and utility.

We present the characteristics (Figure 11) by considering three use cases outlined in Table 4. To facilitate understanding, a simplified example of an online loan application system that leverages an AI model with XAI capabilities is considered. The system uses seven input features where salary, net worth, and age, are protected features that require privacy preservation. The use cases describe the following scenarios:

- Use Case 1 considers intentional privacy leakage through an adversary.

- Use Case 2 involves interaction of a layman end-user, i.e., a bank's customer, with the XAI system. The end-user is provided an explanation of an automated decision directly through the system and indirectly through a human. Let's assume that in this use case, the loan was denied because the applicant salary was < 40K and age was > 50 years.

- Use Case 3 considers the interaction of technical support, i.e., AI developer and quality engineer, with the XAI system.

**Table 4** Use cases for privacy preserving XAI in an online loan application system.

| Property | Details |
|---|---|
| System | Online loan application system |
| Model owner | Bank |
| Model input features | salary, net worth, age, length of credit history, occupation, working hours per week, education |
| Sensitive features | salary, net worth, age |
| Use Case 1 | Adversary with black-box access to the system. |
| Actors | Adversary |
| Overview | An adversary secures black-box access to the bank's model through the online application system. The adversary attempts different queries and observes the outputs generated by the system. |
| Query data | (i) randomly generated queries.
(ii) targeted queries using prior information. |
| Use Case 2 | Customer accessing explanation of the application outcome. |
| Actors | Customer, bank executive |
| Overview | A customer submits an online application for a loan and is given a denied result. The customer is provided with:
(i) an automated explanation.
(ii) a consultation with a bank executive to discuss the result. |
| Query data | salary = $35K, net worth = $75K, age = 55 years, length of credit history = 30 years, occupation = office executive, working hours per week = 25, education = diploma. |
| Use Case 3 | AI developer accessing explanation for troubleshooting a reported error case and a quality engineer subsequently validating the system updates. |
| Actors | AI developer, quality engineer |
| Overview | An error is reported on a specific query and a developer updates the model during debugging. The developer accesses the explanation of the error case to verify the results. Finally, a quality engineer validates the system updates with another round of testing. |
| Query data | Synthetic query similar to the error case requiring troubleshooting. |

We propose ten characteristics of privacy preserving XAI that aim to balance privacy, explainability and utility. The first six characteristics are derived from privacy attacks and unintentional leakage discussed in Section 4. The remaining four characteristics are focussed on addressing performance issues and ensuring regulatory compliance. The proposed characteristics are as follows:

## 6.1   Prevent training data identification

XAI tools should be designed such that the identification of individuals used in training the model is not facilitated through them. In Use Case 1, if the adversary has access to a specific individual's input details and retrieves the corresponding outputs including the outcome and explanations, no additional advantage should be provided through explanations in determining if the individual was used in training the bank's model. Thus, the explanations should be resilient to membership inference (Section 4.2.1).

## 6.2   Prevent sensitive data inference

XAI tools should be designed to prevent reconstruction or inference of sensitive attributes of individuals. In Use Case 1, if the adversary has access to the non-sensitive features of an individual and the outcome of the loan application but is unaware of any sensitive feature such as salary or age, the explanations provided should not aid in inferring these sensitive features of the individual. Thus, the explanations should be resilient to model inversion (Section 4.2.2).

## 6.3   Prevent reverse engineering of model

XAI tools should be designed to prevent reverse engineering of the model functionalities. In Use Case 1, the adversary, by querying the bank's model and by inspecting the explanations, should be unable to build a surrogate of the original model. Thus, the explanations should be resilient to model extraction (Section 4.2.3).

## 6.4   Prevent property inference

XAI tools should be designed to prevent the inferencing of aggregate properties of the training data. In Use Case 1, by using targeted queries on the bank's model, the adversary should be unable to exploit explanations in determining group properties such as the ratio of old and young customers in the training data or the distribution of wealthy and average income training participants. Thus, the explanations should be resilient to property inference (Section 4.2.4).

## 6.5   Prevent direct exposure

Explanations generated by XAI tools should not disclose personally identifiable and/or sensitive information to unauthorized individuals (Chen et al., 2022a). In Use Case 2, when the customer seeks an explanation on his/her application outcome, the explanation might indicate the failure to meet respective thresholds of $40K for salary and 50 years for age. Revealing actual values of protected features would breach the customer's privacy when accessed by other actors, such as the bank executive during the customer's consultation. The customer may, however, subsequently provide consent to the executive to retrieve their personal and financial details from the bank's records for consultation.

## 6.6   Prevent indirect exposure

The content of the provided explanations should not indirectly expose personally identifiable and/or sensitive information through correlated or proxy features to unauthorized individuals. In Use Case 2, if the explanation discloses a non-sensitive attribute such as the length of credit history, to actors other than the customer, it could indirectly lead to exposure of a sensitive attribute, such as age, due to the strong correlation between the two attributes.

## 6.7   Access control of explanations

The content of explanations should be accessible only to authorized users (Blanco-Justicia et al., 2020; Kuppa & Le-Khac, 2021) with details provided on need-to-know basis. In Use Case 2, the customer is authorized to access his/her own explanation as it pertains to their specific application outcome. The bank executive should be authorized to access the explanation only if a human intermediary is required to enhance the

process of explanation for the customer. In Use Case 3, the AI developer and quality engineer should be permitted to access explanations and outcomes only for synthetic queries generated to simulate specific error cases rather than for real production data.

### 6.8 Upholding of explanation quality

The quality of explanations should not be compromised by the introduction of privacy preservation measures. Explanations must remain useful and meaningful (Shokri et al., 2021) to target stakeholders. In Use Cases 2 and 3, the details contained in the explanations to respective users should assist them in completing their tasks effectively and/or help them interpret the outcome of the AI system.

### 6.9 Acceptable run time

The run time of XAI methods, being an important evaluation metric (Bodria et al., 2023), should not deteriorate by introduction of privacy preservation measures. In Use Cases 2 and 3, the explanation recipients should see the outputs within an acceptable timeframe. A long turnaround time may lead to the explanations becoming ineffective for the task at hand.

### 6.10 Compliance with applicable AI/privacy regulations

XAI being an AI system, should comply with applicable AI and privacy regulations specfic to the jurisdiction in which it operates. For instance, if the XAI is deployed in Canada with Canadian residents as the target users, it must adhere to the provisions of the Artificial Intelligence and Data Act (AIDA, 2022). Users should be clearly informed of the XAI capabilities of the system including the types and content of explanations and the third parties with whom the explanations may be shared. Furthermore, appropriate consent must be obtained from users, as required by applicable laws and regulations.

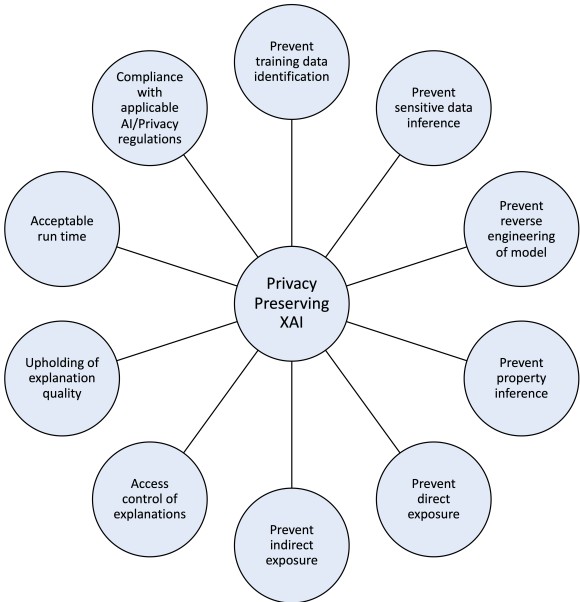

Figure 11: Proposed privacy preserving XAI characteristics.

# 7    Discussion

In this section, we summarize the results of our work, open issues, and challenges in the field. Additionally, we offer recommendations for future research directions to advance the development and deployment of privacy preserving XAI systems.

## 7.1    Summary and implications

The comprehensive review of existing literature facilitated a synthesis of current knowledge on the conflict between privacy and explainability, both being important pillars of Trustworthy AI. The analyzed studies demonstrate that the additional information provided in the form of explanations can benefit adversaries in launching privacy attacks in XAI. We identified and categorized certain types of privacy leakage due to malicious intent of adversaries as intentional causes. These include membership inference, model inversion and model extraction, all of which are demonstrated on explanations generated using different methods. These attacks pose a threat to the privacy of individuals whose data is contained in the training set, thus increasing the risk of identification of individuals or exposure of their sensitive information. Moreover, reconstruction of data from inference-time queries renders active XAI users vulnerable to similar privacy breaches (Zhao et al., 2023; 2021). The threat of model extraction through explanations, targets the confidentiality and intellectual property of model owners. While property inference can expose sensitive aggregates or group properties of the training data, our review found no evidence of such attacks targeting XAI systems specifically. Beyond privacy attacks, ML models exhibit inherent privacy vulnerabilities, such as memorization of training data or overfitting, which can lead to various inferencing attacks. These privacy problems are inherited in XAI systems, and we have categorized them as unintentional causes. Additionally, the explanation content can be at direct threat of privacy breaches by unauthorized users due to lack of access control, or through proxies and correlated fields. Such vectors further compound the privacy risks associated with deploying XAI technologies in sensitive domains.

Due to the growing concerns surrounding privacy risks of explainability, researchers have proposed defense mechanisms for privacy preservation with explanations. This review identifies that techniques, such as DP and anonymization, are extensively explored in this context, as evidenced by the number of studies that have employed these methods. However, there remains limited exploration of alternative approaches, including knowledge integration, cryptography, and perturbation, all of which present promising avenues for enhancing privacy preservation. Hence there is scope to utilize these underexplored techniques to achieve privacy in XAI systems. In distributed environments, Fed-XAI attempts to achieve explainability while preserving privacy of local data, yet its current implementations are insufficient to guarantee privacy in the generated explanations.

The investigation of privacy risks and preservation methods in XAI has led to the identification of key characteristics that privacy preserving explanations must exhibit. In addition to being resilient to privacy attacks and preventing both direct and indirect exposure of sensitive information, explanations should satisfy performance and utility constraints. Given that explanations may contain potentially identifiable data and may be subject to legal and regulatory frameworks, they are required to comply with the applicable AI and privacy laws within the jurisdiction. This article identifies and highlights a gap in the research of methods within the field of XAI, i.e., explainability methods should be designed considering privacy as a system requirement. The findings of this paper can be utilized by researchers to understand state-of-the-art privacy attacks and corresponding preservation methods. Practitioners can leverage these insights to enhance their understanding of the privacy risks associated with XAI and identify potential solutions to mitigate those risks across various XAI methods.

## 7.2    Design considerations for privacy preserving XAI

Many unintentional leaks identified in this review can be avoided through responsible design practices. Safeguards in data handling such as data minimisation, deidentification and anonymisation of training data, or the use of synthetic data for training models when sensitive or personally identifiable information is involved, can help minimize the risks. During training, regularisation techniques can prevent overfitting and

memorisation. The use of private training algorithms, such as differentially private training, can provide a privacy guarantee against leakages. At the output interface, the explanation API can include role-based access of explanation content to restrict the availability of sensitive information to end-users on a need-to-know basis. The use of logging at output for monitoring, tracing and privacy audits can further aid the accountability of the system.

Steps can also be taken to prevent intentional leaks by restricting explanation APIs to authorised users. Restraining unlimited query access and setting the number of queries depending on users' roles can prevent misuse of explanation APIs by adversaries. Query monitoring can further help detection of anomalies. In addition to securing the interfaces, model owners should give due consideration to the type of access provided to users. When the model is available through an API, black-box privacy attacks are possible. However, when the model as a whole is released to users, model owners have no control on its usage and white-box privacy attacks become possible, thus giving additional avenues to adversaries to penetrate the system. Developers can use documented XAI privacy attacks for testing of systems during development for evaluation of the risk of privacy breaches. These evaluations can assist in making decisions about integration of appropriate privacy preservation methods as needed.

## 7.3 Open issues and challenges

Based on the privacy risks and mitigation methods surveyed, key open issues and challenges have been identified that require further attention. These challenges underscore the complexity of balancing privacy with the need for explainability in AI systems. In particular, the following issues remain critical:

- **Lack of user-centricity in development of privacy preserving XAI:** End-users are an integral and inseparable component of XAI as they directly engage and draw insights from the content generated by these systems. While current XAI methods are predominantly model-centric, focussing on model development, evaluation and audit processes (Kaplan et al., 2024), there is an increasing need for a shift towards a user-centric approach. When considering privacy as a system requirement or using existing privacy preserving solutions with explainability methods, human users are absent from the design and development of these approaches.

- **Lack of standardised approach for privacy evaluation of XAI:** Existing XAI literature provides evaluation metrics to assess aspects such as faithfulness, complexity, localisation, randomisation and robustness (Hedström et al., 2023). However, though privacy is fundamental in Trustworthy AI, currently there is a lack of standardised approach to evaluate the privacy of explanations. The lack of suitable quantitative metrics in this area prevents an evaluaton of safety with respect to privacy or comparison between methods for selection of those that satisfy the safety requirements of an application domain.

- **Trade-offs in privacy, explainability and utility:** The introduction of privacy enhancing technologies often result in adverse effects on model utility (Ezzeddine et al., 2024; Harder et al., 2020) and the quality of explanations. For instance, the perturbation of classifier weights of support vector machines for privacy preservation is observed to deteriorate the classification accuracy and credibility of counterfactuals (Mochaourab et al., 2023). Similarly, the use of differential privacy in neural network models is found to lower its accuracy (Blanco-Justicia et al., 2023) and explanation quality (Liu et al., 2024). Thus when targeting these 3 properties simultaneously in an XAI system, there are often trade-offs involved.

- **Lack of XAI methods that are privacy preserving by design:** As emphasized by Hoepman (2014), privacy is a core property of computer systems that requires addressing from system design phase, rather than treated as an add-on. However, current explainability approaches do not consider privacy as a system requirement. Due to the overlook of this safety aspect, design flaws may be introduced which can lead to unintentional privacy leakage. Adversaries may exploit these flaws to cause intentional leakage.

- **Lack of privacy preserving XAI for Gen-AI and LLMs:** XAI research has mainly focused on discriminative models that produce decision boundaries, and there has been limited work on developing explainability methods for Gen-AI and LLMs (Schneider, 2024b; Sun et al., 2022; Weisz et al., 2023). Due to the complex structure and vast number of parameters in these models, traditional explainability methods become impractical to them (Zhao et al., 2024). These models have privacy issues, such as memorization of training data that escalates as the models become larger (Carlini et al., 2021). In addition, downstream private datasets used for in-context learning in LLMs, are found to be susceptible to membership inference (Duan et al., 2023). When creating explainability approaches for these systems, privacy should be considered as an integral requirement.

### 7.4 Recommendations for future work

Based on the open issues and challenges outlined in the previous subsection, we have the following recommendations for future research in this area:

- **Improving usability of privacy preserving XAI:** Explanations should be designed to meet the diverse information needs of users and integrate user-centric design principles (Ali et al., 2023; Kaplan et al., 2024). The delivery of explanations is required to be in a format that is accessible and meaningful, taking into consideration the varying levels of expertise and requirements of different user groups. This transition would prioritize providing need-to-know information to end-users based on their specific roles within the system. Furthermore, to enhance the effectiveness of explanations, appropriate tools such as interactivity and visualization should be used to enhance the process of explanation and deepen users' understanding (Bo et al., 2024). In addition, application of the 3C-principle of context, content, and consent (Brunotte et al., 2023) can improve the usability of XAI tools by satisfying their requirements and expectations.

- **Development of a standardised approach of privacy evaluation of XAI methods:** The development of a standardised approach to measure the leakage of privacy through explanations, will benefit developers in gauging the privacy of specific explanations so that appropriate preservation techniques can be integrated as needed. The development of quantitative privacy metrics can facilitate comparison between methods for adoption of those that are privacy safe and suitable for use in specific domains depending on the regulation requirement or the risk involved.

- **Balancing trade-off in privacy, explainability and utility:** Determining the appropriate trade-off between the triad of privacy, explainability and utility, can help to achieve the right measure of balance between these properties. This could be achieved with the help of tools or tuning mechanisms. For instance, the use of compatibility matrix (Abbasi et al., 2024) or hyperparameters, such as the privacy budget $\epsilon$ in differential privacy or the parameter $k$ in k-anonymity, is useful in tuning the desired level of these required properties. Similar tuning mechanisms can be integrated in other privacy preserving approaches to achieve the required trade-off. Metrics, such as trade-off score (Abbasi et al., 2024), could be useful to quantify and monitor the balance of these properties enabling researchers and practitioners to adjust the parameters based on the trade-offs involved.

- **Examining and improving trade-off in different privacy preserving methods for XAI:** Different privacy preservation methods applicable to XAI are discussed in Section 5. Analysing the privacy-explainability-utility trade-off of these methods will identify effective solutions and their limitations. While techniques, such as differential privacy and anonymization, have been mainly explored in XAI systems, other underutilized techniques such as use of knowledge integration and cryptographic protocols, could provide alternative approaches. Distributed privacy enhancing solutions, such as Fed-XAI, warrant further investigations to determine strategies to mitigate possible privacy leakages from XAI components. By systematically examining and comparing these various privacy preserving techniques, researchers can identify best practices and design hybrid approaches to effectively balance different properties.

- **Development of XAI methods that are privacy preserving by design:** The characteristics of privacy preserving XAI outlined in this review, can aid researchers and developers in building

explainability methods that are privacy-enhanced by design (Bozorgpanah & Torra, 2024). Furthermore, there is growing interest in neuro-symbolic approaches (Hitzler et al., 2022) and semantic technologies (Seeliger et al., 2019) as potential solutions as explainable-by-design strategies. Researchers can investigate the privacy safety of these techniques and integrate approaches to prioritize privacy.

- **Privacy preserving XAI for Gen-AI and LLMs:** XAI plays a vital role in fostering trustworthiness (Wang & Ding, 2024) and ensuring ethical applications of Gen-AI and LLMs (Luo & Specia, 2024). However, as explainability is introduced in these models, it is important to ensure that it does not exacerbate the inherent privacy issues of these models or create new vulnerabilities. A privacy analysis of explainability methods in the early stages of development and the use of privacy attacks for auditing (Carlini et al., 2021), can boost the development of privacy-enhanced systems. Recently, Chain-of-Thought (CoT) (Wei et al., 2022) prompting has gained traction in eliciting the step-by-step reasoning of models and providing self-explanations in the process. However, the interaction with untrusted LLMs can be a threat to users' privacy, hence future work can be directed towards privacy preserved mechanisms (Bae et al., 2025) in these models. Methods such as retrieval-augmented generation (RAG) for fine tuning of outputs by augmenting external data sources (Zeng et al., 2024) and mechanistic interpretability (Section 2.3.8) can also be investigated as possible solutions towards achieving privacy preservation XAI. Thus, with the growing accessibility and widespread use of Gen-AI and LLMs, developing appropriate user-centric privacy preserving explainability techniques is an important avenue for further research.

- **Evaluation of privacy preserving XAI characteristics:** The characteristics of privacy preserving XAI that we propose, aims to highlight the desirable qualities that XAI should exhibit to protect privacy while producing useful explanations to the target users. In further work, we will evaluate current XAI methods in light of these proposed characteristics to determine gaps in the methods and inform strategies for improvement. We will also enhance current methods so that the generated explanations better align with the proposed characteristics. We aim to improve the applicability of the characteristics through the evaluation of XAI methods.

## 8 Conclusion

XAI is an active field of research and a crucial pillar of Trustworthy AI. It aims to bring logical explanations, a fundamental property of all computer systems, to black-box AI models. Explainability of models is essential to secure user trust in automated outcomes, especially in critical domains where such outcomes have high impact on the lives of individuals. Though explainability has emerged as a gold standard for Trustworthy AI, previous works have highlighted potential privacy risks of introducing transparency to black boxes. To the best of our knowledge, there is a lack of detailed review on the tension between privacy and explainability. In this article, we have focused on this gap and conducted a scoping review to elicit details on the privacy risks posed by XAI and the corresponding solutions for privacy preservation in XAI. Our review draws attention to the intentional and unintentional misuse of explanation interfaces and the pressing need for developing XAI that is privacy preserving. In addition to reviewing the privacy risks and the progress achieved by researchers in achieving privacy improvement in XAI systems, we propose the characteristics of privacy preserving XAI, to assist AI engineers and researchers in understanding the requirements of XAI that achieves privacy with utility. We base these characteristics on the identified risks, the encountered performance issues, and the expected regulatory compliance. The characteristics can be utilized for designing new explainability methods and for evaluation of existing methods. Finally, we conclude the article by identifying the open issues and challenges in the field and provide recommendations for future work. Among the directions identified, developing privacy metrics, creating privacy preserving explanations for generative models and balancing the trade-off of privacy, utility and explainability in existing and new XAI methods, will determine its success as a foundation pillar of Trustworthy AI.

**Broader Impact Statement**

This research was conducted to determine if two pillars of Trustworthy AI, namely, privacy and explainability, can co-exist. The privacy risks highlighted here have been consolidated from current literature to create awareness when using explainable methods in bringing transparency to black-boxes.

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
