# OpenReview forum: "Privacy Risks and Preservation Methods in Explainable Artificial Intelligence: A Scoping Review"
_TMLR — Accepted by TMLR_

### Review · Reviewer_dAeN · 2025-07-25

**Summary Of Contributions:**

This paper presents a scoping review on privacy risks and preservation methods in Explainable AI (XAI), driven by the growing regulatory and practical demand for transparent, high-stakes AI systems. It categorizes privacy risks, surveys existing privacy-preserving techniques, and proposes key characteristics for trustworthy, privacy-aware explanations. By synthesizing current knowledge on explainability and privacy, the work outlines major attack and defense strategies and provides practical recommendations for developing secure and reliable XAI systems.

**Audience:**

Yes

**Audience Explanation:**

Yes, this work made a comprehensive survey that synthesizes current knowledge on privacy risks and preservation in Explainable AI.

**Claims And Evidence:**

Yes

**Claims Explanation:**

Yes, the claims made in the submission are supported by accurate, convincing, and clear evidence.

**Requested Changes:**

### Strengths
- The survey is well-structured and comprehensive, using a systematic search strategy to provide an overview of privacy risks and preservation methods in XAI.
- The paper clearly recognizes the trade-off between explainability and privacy, highlighting key challenges in responsible XAI deployment.
- Visuals and tables clearly summarize research trends, attacks, and defenses for quick understanding.

### Weaknesses
- While figures show numerical distributions of attacks and methods, they lack summary of their empirical effectiveness or real-world impact, making it hard to assess which approaches are most promising.
- Many figures are too simple and lack visual appeal.
- The survey may omit several relevant survey or benchmark studies on privacy risks and preservation in XAI: [1-3]

### Requested Changes
- Include serval key surveys to strengthen literature coverage.
- Update XAI categorization to include recent developments of LLMs.
- Add discussion of open research directions for underexplored risks to guide future work.

Overall, this paper provides a timely and well-structured survey on privacy in XAI, with clear taxonomy and visuals. However, it misses serval relevant works, lacks depth in comparing defenses and privacy-utility tradeoffs, and has limited visual appeal, reducing its engagement for readers. The contribution is solid but incremental, and would benefit from better coverage, analysis, and more compelling presentation.

[1] A Survey of Privacy-Preserving Model Explanations: Privacy Risks, Attacks, and Countermeasures

[2] Privacy-Preserving Explainable AI: A Survey

[3] Privacy Meets Explainability: A Comprehensive Impact Benchmark.

---

> ### Author Response · Authors · 2025-09-11
> **Response to Reviewer dAeN**
>
> Thank you for your insightful comments. We have posted the revised version incorporating your requested changes. Please see our response below to your comments:
>
> > While figures show numerical distributions of attacks and methods, they lack summary of their empirical effectiveness or real-world impact, making it hard to assess which approaches are most promising.
> > Many figures are too simple and lack visual appeal.
>
> We have updated the figures. Figs. 5 to 9 are modified to focus on the adversarial model and its real-world impact due to adversaries. The goal of Fig. 10 is to consolidate and report the defense approaches from current literature without recommending specific approaches as more promising than others.
>
> > The survey may omit several relevant survey or benchmark studies on privacy risks and preservation in XAI: [1-3]
> > Include serval key surveys to strengthen literature coverage.
>
> We have added discussion of [2] in Section 2.4 (please note that [1] is the preprint version of [2]). We decided to exclude [3] since it is a preprint, but we have incorporated your feedback on strengthening literature coverage and included the following additional reviews in Section 2.4:
>
> [4] Hubert Baniecki and Przemyslaw Biecek. Adversarial attacks and defenses in explainable artificial intelligence: A survey. Information Fusion, 107:102303, 2024. ISSN 1566-2535. doi: https://doi.org/10.1016/j.inﬀus.2024.102303. URL https://www.sciencedirect.com/science/article/pii/S1566253524000812.
>
> [5] Maximilian Noppel and Christian Wressnegger. A brief systematization of explanation-aware attacks. In Andreas Hotho and Sebastian Rudolph (eds.), KI 2024: Advances in Artificial Intelligence, pp. 350–354, Cham, 2024b. Springer Nature Switzerland. ISBN 978-3-031-70893-0.
>
> [6] Maximilian Noppel and Christian Wressnegger. Sok: Explainable machine learning in adversarial environments. In 2024 IEEE Symposium on Security and Privacy (SP), pp. 2441–2459, 2024a. doi: 10.1109/SP54263.2024.00021.
>
> [7] Christoforos N. Spartalis, Theodoros Semertzidis, and Petros Daras. Balancing xai with privacy and security considerations. In Computer Security. ESORICS 2023 International Workshops: CPS4CIP, ADIoT, SecAssure, WASP, TAURIN, PriST-AI, and SECAI, The Hague, The Netherlands, September 25–29, 2023, Revised Selected Papers, Part II, pp. 111–124, Berlin, Heidelberg, 2023. Springer-Verlag. ISBN 978-3-031-
> 54128-5. doi: 10.1007/978-3-031-54129-2_7. URL https://doi.org/10.1007/978-3-031-54129-2_7.
>
> > Update XAI categorization to include recent developments of LLMs.
>
> Categories more specific to Gen-AI and LLMs are added in Sections 2.3.5 to 2.3.8.
>
> > Add discussion of open research directions for underexplored risks to guide future work.
>
> We have updated Section 7 to discuss open issues in Section 7.3 and directions for future work in Section 7.4.

---

> > ### Comment · Reviewer_dAeN · 2025-09-18
> >
> > Thank you for addressing my concerns, most of them have been resolved with your updates.

---

### Review · Reviewer_4h3R · 2025-08-20

**Summary Of Contributions:**

This paper presents a scoping review of the current landscape of privacy risks and defenses in explainable AI (XAI). The authors introduce a taxonomy of privacy risks in XAI, provide an overview of existing defenses, and propose desiderata for effective defenses to achieve better privacy-utility trade-offs.

The topic is timely and addresses a key challenge in enabling the real-world deployment of XAI systems. The paper is well-motivated and generally well-written. The main weakness lies in the limited depth of the scoping review and the proposed taxonomy, as well as the insufficient engagement with existing surveys on the topic (see detailed comments below).

**Audience:**

Yes

**Audience Explanation:**

Yes, the paper could be a good reference for anyone interested in learning more about adversarial XAI.

**Broader Impact Concerns:**

Not really applicable. However, the authors could add a section after the conclusion to reiterate the objective of the paper, namely to raise awareness about the privacy risks of XAI.

**Claims And Evidence:**

Yes

**Claims Explanation:**

Yes (with some reserve).

My primary concern relates to the paper’s methodological claim of conducting a scoping review. Specifically, the description of the data charting process is unclear (cf. item 10 of the PRISMA-ScR checklist). Without this clarification, it is difficult to assess the rigor of the review.

In addition, the link between the scoping review methodology and the proposed taxonomy is not well articulated. For instance, the basis for the intentional/unintentional dichotomy is not sufficiently justified. This distinction also appears somewhat misleading, since several phenomena categorized as “unintentional” (e.g., overfitting, memorization) are frequently described in the literature as enabling factors for the attacks classified under the “intentional” category (e.g., membership inference, data extraction).

**Requested Changes:**

- Clarify the exclusion criteria in section 3.1
- Update Section 2.4 to include recent/similar contributions such asref listed below. Include how this work differs overlaps.

	- Noppel, M., & Wressnegger, C. (2024, August). A Brief Systematization of Explanation-Aware Attacks. In German Conference on Artificial Intelligence (Künstliche Intelligenz) (pp. 350-354). Cham: Springer Nature Switzerland.

	- Nguyen, T. T., Huynh, T. T., Ren, Z., Nguyen, T. T., Nguyen, P. L., Yin, H., & Nguyen, Q. V. H. (2024). A survey of privacy-preserving model explanations: Privacy risks, attacks, and countermeasures. arXiv preprint arXiv:2404.00673.

	- Baniecki, H., & Biecek, P. (2024). Adversarial attacks and defenses in explainable artificial intelligence: A survey. Information Fusion, 107, 102303.

- For a reader not familiar with adversarial ML, the paper can be hard to parse. The adversary model (e.g., white-box/black-box access) is not clearly explain. I suggest to have a section where the different aspects of the adversary model are explained.
- Consistently present the attacks through the adversary model, and (minimally) the gain obtained with the explanation to highlight how explanation introduce new risk.

---

> ### Author Response · Authors · 2025-09-11
> **Response to Reviewer 4h3R**
>
> Thank you for your helpful comments. We have updated the manuscript in response to your feedback. Please see our response below to your comments:
>
> > My primary concern relates to the paper’s methodological claim of conducting a scoping review. Specifically, the description of the data charting process is unclear (cf. item 10 of the PRISMA-ScR checklist). Without this clarification, it is difficult to assess the rigor of the review.
>
> We have updated Section 3.1 to include more details on the data charting process. The process of formulating the search criteria with the help of a librarian and the verification and discussion between researchers during the steps of eligibility and extraction is updated.
>
> > the link between the scoping review methodology and the proposed taxonomy is not well articulated. For instance, the basis for the intentional/unintentional dichotomy is not sufficiently justified. This distinction also appears somewhat misleading, since several phenomena categorized as “unintentional” (e.g., overfitting, memorization) are frequently described in the literature as enabling factors for the attacks classified under the “intentional” category (e.g., membership inference, data extraction).
>
> We have added a Section 3.2 to explain the link between the extracted studies and the proposed taxonomy. The term intentional is used to refer to the malicious intent of adversaries while unintentional primarily refers to inadvertent leaks through legitimate users. Please also see the updated Fig. 5 for the respective surfaces where these leaks become possible. We also note in Section 3.2 that the phenomena leading to unintentional leaks may be further exploited for intentional leaks once an adversary is introduced in the system. Current literature also alternatively refers to these two categories of leakages as malicious (intentional) and involuntary (unintentional) [1].
>
> [1] Marija Jegorova, Chaitanya Kaul, Charlie Mayor, Alison Q. O’Neil, Alexander Weir, Roderick Murray-
> Smith, and Sotirios A. Tsaftaris. Survey: Leakage and Privacy at Inference Time. IEEE Transactions on
> Pattern Analysis and Machine Intelligence, pp. 1–20, 2022. ISSN 0162-8828, 2160-9292, 1939-3539. doi:
> 10.1109/TPAMI.2022.3229593. URL https://ieeexplore.ieee.org/document/9987657/.
>
> > Clarify the exclusion criteria in section 3.1
>
> Section 3.1 is updated to clarify the exclusion criteria for preprints, surveys and studies on adversarial XAI without overlap on privacy.
>
> > Update Section 2.4 to include recent/similar contributions such as ref listed below. Include how this work differs overlaps.
>
> Section 2.4 is updated to include reviews [2-4] and compare with our review as suggested. In addition, we have also added [5-6] to strengthen literature coverage.
>
> References:
> [2] Noppel, M., & Wressnegger, C. (2024, August). A Brief Systematization of Explanation-Aware Attacks. In German Conference on Artificial Intelligence (Künstliche Intelligenz) (pp. 350-354). Cham: Springer Nature Switzerland.
>
> [3] Nguyen, T. T., Huynh, T. T., Ren, Z., Nguyen, T. T., Nguyen, P. L., Yin, H., & Nguyen, Q. V. H. (2024). A survey of privacy-preserving model explanations: Privacy risks, attacks, and countermeasures. arXiv preprint arXiv:2404.00673.
>
> [4] Baniecki, H., & Biecek, P. (2024). Adversarial attacks and defenses in explainable artificial intelligence: A survey. Information Fusion, 107, 102303.
>
> [5] Maximilian Noppel and Christian Wressnegger. Sok: Explainable machine learning in adversarial environments. In 2024 IEEE Symposium on Security and Privacy (SP), pp. 2441–2459, 2024a. doi: 10.1109/SP54263.2024.00021.
>
> [6] Christoforos N. Spartalis, Theodoros Semertzidis, and Petros Daras. Balancing xai with privacy and security considerations. In Computer Security. ESORICS 2023 International Workshops: CPS4CIP, ADIoT, SecAssure, WASP, TAURIN, PriST-AI, and SECAI, The Hague, The Netherlands, September 25–29, 2023, Revised Selected Papers, Part II, pp. 111–124, Berlin, Heidelberg, 2023. Springer-Verlag. ISBN 978-3-031-
> 54128-5. doi: 10.1007/978-3-031-54129-2_7. URL https://doi.org/10.1007/978-3-031-54129-2_7.
>
> > For a reader not familiar with adversarial ML, the paper can be hard to parse. The adversary model (e.g., white-box/black-box access) is not clearly explain. I suggest to have a section where the different aspects of the adversary model are explained.
>
> We have included Section 4.1 to explain the threat model of XAI and updated Fig. 5 accordingly.
>
> > Consistently present the attacks through the adversary model, and (minimally) the gain obtained with the explanation to highlight how explanation introduce new risk.
>
> Sections 4.2.1 to 4.2.4 are updated to present the attacks through the adversarial model. Figs. 6 to 9 are accordingly updated. Expressions (1) to (8) are introduced for readers to understand the main difference between the attacks with and without XAI.
>
> > Update broader impact statement
>
> This section is added

---

> > ### Comment · Reviewer_4h3R · 2025-09-18
> > **Response to authors**
> >
> > The revised manuscript and your responses have addressed my comments. Thank you!

---

### Review · Reviewer_ULTM · 2025-08-22

**Summary Of Contributions:**

The work looks at the direct impact of enhancing model utility using eXplainable AI (XAI) on the privacy leakage of the model. In particular, the authors note interesting progressions in XAI and the upward trend of related privacy discussions in this space. Furthermore, they go over exact references for where XAI methods have now been developed so as to be resilient to privacy concerns. Overall, it seems that the paper aims to address the gap of explaining the correlation between XAI, privacy and ways to mitigate any possible privacy sensitive outputs.

**Additional Comments:**

N/A

**Audience:**

Yes

**Audience Explanation:**

The author's work, as currently presented, has the following direct benefits,
- Provides a comprehensive list of references for understanding tangible and exact privacy risks attached to XAI methods (Section 4).
- Enables a better understanding of ways to limit such risks (Section 5).

**Claims And Evidence:**

Yes

**Claims Explanation:**

Given the original premise of the paper, about directly connecting XAI and its impact on privacy, the authors successfully show that, especially in Section 4. Furthermore, in Section 5, the authors discuss remedial techniques for lowering the impact of XAI-related privacy impact.

**Requested Changes:**

Although the risks discussed in the work are valid and warrant discussion, it seems that the survey paper might be too broad rather than a targeted delivery of the most important items. I would recommend the following changes,

- Focus specifically on a risk-based segregation, such as the biggest difference in models that include XAI and ones that do not.
- Presently, it's unclear without reading a lot of content what the next biggest challenges are in this space. I would encourage the authors to add the same.
- Have more regular summaries or concluding remarks that can help readers understand items of interest more clearly, regularly, and possibly exclude or limit the discussion of low-risk XAI-privacy items altogether for brevity.
- Focus more on the design approach of XAI systems, which may be the crux of reducing the privacy leakage. Would love more thorough discussions about the same.

Overall, it seems that a detailed discussion about the highest risks posed by XAI might benefit a reader much more than a broad general XAI-privacy connection.

---

> ### Author Response · Authors · 2025-09-11
> **Response to Reviewer ULTM**
>
> Thank you for your valuable comments. We have updated the manuscript in response to your feedback. Please see below for the details on the changes made:
>
> > Focus specifically on a risk-based segregation, such as the biggest difference in models that include XAI and ones that do not.
>
> The threat model of XAI is explained in a new Section 4.1. Expressions (1) to (8) in Sections 4.2.1 to 4.2.4 are introduced to describe the primary difference in the adversarial models with and without XAI.
>
> > Presently, it's unclear without reading a lot of content what the next biggest challenges are in this space. I would encourage the authors to add the same.
>
> We have streamlined Section 7 to include open issues in Section 7.3 and future research directions in Section 7.4. We hope that this will enhance readability of the section.
>
> > Have more regular summaries or concluding remarks that can help readers understand items of interest more clearly, regularly, and possibly exclude or limit the discussion of low-risk XAI-privacy items altogether for brevity.
>
> Sections 2.3, 4.2 and 7 are streamlined to help readability. As suggested, low risk privacy items are removed from the discussion.
>
> > Focus more on the design approach of XAI systems, which may be the crux of reducing the privacy leakage. Would love more thorough discussions about the same.
>
> We have introduced a new Section 7.2 to discuss the design approach of XAI to target reduction of both intentional and unintentional leaks.

---

### Decision · Action_Editor_RWTN · 2025-10-20

**Recommendation:** Accept as is

**Additional Comments:**

Reviewers agree that the survey covers the timely and important topic of XAI and privacy.
The authors have appropriately addressed the reviewers' suggestions, which focused on updating the survey methodology and comparison to related work.
Overall, the reviewers recommend acceptance, seeing the paper as a comprehensive survey that synthesizes current knowledge on privacy risks and preservation in XAI.

**Audience:**

Yes

**Audience Explanation:**

XAI and privacy are key research topics in machine learning and are of interest to TMLR readers.

**Claims And Evidence:**

Yes

**Claims Explanation:**

The paper surveys the relationship between XAI and privacy, focusing on risks and defense methods.
The authors introduce a taxonomy of privacy risks, review existing risks and defenses, and propose desiderata for better privacy-utility trade-offs.